# Low-intensity pulsed ultrasound stimulation (LIPUS) modulates microglial activation following intracortical microelectrode implantation

Fan Li[1,2,3,12], Jazlyn Gallego[1,2,12], Natasha N. Tirko[4], Jenna Greaser[5], Derek Bashe[6], Rudra Patel[7], Eric Shaker[1], Grace E. Van Valkenburg[1], Alanoud S. Alsubhi[5], Steven Wellman[8], Vanshika Singh[1], Camila Garcia Padilla[1,2], Kyle W. Gheres[5], John I. Broussard[5], Roger Bagwell[5], Maureen Mulvihill[5] & Takashi D. Y. Kozai[1,2,9,10,11] ✉

Microglia are important players in surveillance and repair of the brain. Implanting an electrode into the cortex activates microglia, produces an inflammatory cascade, triggers the foreign body response, and opens the blood-brain barrier. These changes can impede intracortical brain-computer interfaces performance. Using two-photon imaging of implanted microelectrodes, we test the hypothesis that low-intensity pulsed ultrasound stimulation can reduce microglia-mediated neuroinflammation following the implantation of microelectrodes. In the first week of treatment, we found that low-intensity pulsed ultrasound stimulation increased microglia migration speed by 128%, enhanced microglia expansion area by 109%, and a reduction in microglial activation by 17%, indicating improved tissue healing and surveillance. Microglial coverage of the microelectrode was reduced by 50% and astrocytic scarring by 36% resulting in an increase in recording performance at chronic time. The data indicate that low-intensity pulsed ultrasound stimulation helps reduce the foreign body response around chronic intracortical microelectrodes.

Inflammation plays a critical role in the brain's defense against diseases and damage[1]. Although acute inflammation helps to clear cellular debris, chronic inflammation may exacerbate tissue damage[2]. Chronic inflammation impairs neurovascular coupling[3], disrupts the energy supply to the brain, and elevates oxidative stress[4]. Furthermore, neuroinflammation increases the permeability of the blood–brain barrier (BBB)[5], allowing infiltration of blood contents such as fibrinogen, which can lead to damage in brain tissue[6,7]. These changes induced by inflammation are linked to neurodegenerative diseases such as Alzheimer's disease[8], multiple sclerosis[9], and Parkinson's disease[10], as well as play a role in the aging process[11], pain sensation[12], gliomas[13], injuries induced by

[1]Department of Bioengineering, University of Pittsburgh, Pittsburgh, PA, USA. [2]Center for Neural Basis of Cognition, Pittsburgh, PA, USA. [3]Computational Modeling and Simulation PhD Program, University of Pittsburgh, Pittsburgh, PA, USA. [4]Department of Biochemistry and Molecular Biology, Pennsylvania State University, University Park, PA, USA. [5]Actuated Medical, Bellefonte, PA, USA. [6]Washington University in St. Louis, St. Louis, MO, USA. [7]Department of Neuroscience, University of Pittsburgh, Pittsburgh, PA, USA. [8]Columbia University, New York, NY, USA. [9]Center for Neuroscience, University of Pittsburgh, Pittsburgh, PA, USA. [10]McGowan Institute of Regenerative Medicine, University of Pittsburgh, Pittsburgh, PA, USA. [11]NeuroTech Center, University of Pittsburgh Brain Institute, Pittsburgh, PA, USA. [12]These authors contributed equally: Fan Li, Jazlyn Gallego. ✉e-mail: TK.Kozai@pitt.edu

stroke[14], traumatic brain injury[15], and the foreign body response (FBR) to brain implants[16].

Penetrating microelectrode arrays that interface with the nervous system are front-end components of brain–computer interfaces (BCI), which have demonstrated remarkable potential for restoring motor and sensory function[17–19]. One key challenge is the complex FBR caused by neuroinflammation after the insertion of microelectrodes. Neuroinflammation is a multifaceted process orchestrated through interactions of blood cells[20], endothelial cells[21], and glial cells[22], particularly microglia[23–26]. These microglial processes can often be detrimental to intracortical microelectrode interfaces, which are designed to detect neuronal signals and study neural activity[23–26]. The implantation of a foreign body such as a microelectrode into the brain disrupts the BBB[27,28], degenerates neurons[16,29,30] and oligodendrocytes[16,31–33], and activates microglia[34–38], astrocytes[34–42], and NG2 glia[39,41,42]. Among glial cells, microglia are first responders and important mediators of neuroinflammation, protecting the brain from injury[38,43]. Microglia's filopodia, or processes, enable them to efficiently survey the surrounding area, detecting pathogens, disturbances, or foreign bodies[38,44]. Previous studies show that microglia direct their processes toward the microelectrode minutes after implantation, followed by astrocyte processes on the order of hours and NG2 glial processes on the order of days[36,39,40,45,46]. During this phase, microglia tend to extend both a longer and greater number of processes toward the injury site while reducing the length and number of processes away from the injury site[38]. Next, microglia begin to migrate toward microelectrodes within 12–24 h, followed by NG2 glia[36,39,40,45,46]. By contrast, astrocytes do not migrate, instead they swell and become hypertrophic[39]. Over weeks, these glial cells form the glial scar surrounding the implanted probe[47], resulting in the characteristic FBR[36,39,40,46].

This FBR adds an insulating layer on the microelectrode[48] increasing the impedance[49] and the recorded noise floor[50] of the microelectrode. Further, persistent microglial activation upregulates production of proinflammatory cytokines[51–53] contributing to progressive neurodegeneration, reducing the number of recorded neurons surrounding the microelectrode[54], and decreasing the number of detectable single-units[55]. Additionally, proinflammatory microglia can attach to blood vessels, initiate upregulation of proinflammatory profiles, and phagocytose astrocyte endfeet, breaking down the neurovascular unit[56]. Phagocytic microglia also contribute to the loss of neurons and synapses via complement activation[57,58], disrupting the neural circuit and adversely affecting the propagation of neural signals. Moreover, proinflammatory microglia promote the release of nitric oxide (NO)[59], causing abnormal dilation of the cerebral vessels[60]. Together with other glial cells, such as astrocytes, NG2 glial cells, and oligodendrocytes[16,32,39,40], this neuroinflammatory response increases the noise in electrophysiological recordings, decreases the strength of the neural signal being recorded, and obstructs the tissue–microelectrode integration[61]. Although administration of drugs such as dexamethasone[37] and HOE-642[34] or coating microelectrodes with zwitterionic polymer[35] and neuroadhesive L1[36] reduces microglial activation, these interventions require either recurrent injections or complex manufacturing processes[62]. Therefore, modulating microglial changes following microelectrode implantation remains a challenge in neuroscience research and clinical BCI applications.

Ultrasound stimulation is an emerging tool for neuromodulation, achieved through the delivery of high-frequency mechanical waves to neural tissue[63–66]. Particularly, low-intensity pulsed ultrasound stimulation (LIPUS) delivers intermittent waves that have been shown to reduce inflammation and promote tissue healing without causing overwhelming thermal effects[67]. Recent studies report that LIPUS can act on mechanosensitive ion channels[68,69] such as piezo1[70] to have a gliomodulatory effect, polarize microglia toward anti-inflammatory phenotypes[71], increase BDNF signaling pathways[66,72,73], reduce proinflammatory cytokine expression in microglia[74], and inhibit the ROCK1/

p-MLC2 signaling pathway[75]. While LIPUS has been demonstrated to have positive cognitive effects in early diseases[67,76], stroke[75,77,78], and brain injury models[79,80], the impact of LIPUS to attenuate a more persistent FBR caused by microelectrode implantation has not been explored. Compared to the administration of drugs and microelectrode coating techniques, LIPUS has the potential to target tissue at the site of injury with high spatial resolution and without the need for complex manufacturing processes[81]. In addition, LIPUS is noninvasive[82], thus minimizes the side-effects such as infusion-related reactions during drug injection procedures, inefficient drug release from electrode housing structures, and delamination between coating materials and microelectrodes[62,83].

In this study, we aimed to investigate the effects of LIPUS on microglial activity, FBR, and the BBB and test whether LIPUS could rescue the degrading recording performance of chronic implants. Using two-photon imaging techniques, we quantified microglial migration, activation, surveillance, encapsulation over microelectrodes, density, and association with blood vessels. Also, we monitored changes in the diameter of cerebral blood vessels surrounding implanted microelectrodes. Furthermore, we tested the hypothesis that LIPUS treatment will reduce microglial activation, decrease microglial coverage on the probe, and reduce the ratio of microglia associated with blood vessels compared to the untreated control group over a period of 28 days following microelectrode implantation. Our results demonstrated that LIPUS treatment effectively enhanced initial microglial migration and surveillance and facilitated the transition of activated microglia back to a ramified state. In addition, LIPUS reduced the extent of microglial coverage on the probe, the ratio of microglia associated with blood vessels, and the dilation of cerebral blood vessels. Finally, LIPUS in rats increased the recording performance of electrodes in L5 as well as reduced astrocyte activation. These findings suggest that LIPUS has the potential to serve as a noninvasive therapeutic intervention for modulating the FBR, thereby facilitating recovery and improving device-tissue integration following microelectrode implantation.

## Results

### Experimental setup for LIPUS stimulation and two-photon imaging

To optimize the application of LIPUS on brain tissue, we conducted in vitro testing of LIPUS parameters (Fig. 1a). The thermal effects of LIPUS were monitored via the implantation of a thermocouple around 1.5 mm deep into the rodent cortex. A power supply voltage of 135 V peak-to-peak for the transducer was carefully selected to ensure that the temperature change was below 1 °C throughout the LIPUS treatment (Fig. 1b). In the in vivo LIPUS group (N = 7), animals were treated with LIPUS followed by the two-photon imaging on day 0, 1, 2, 3, 4, 5, 6, 7, 14, 21, 28 post microelectrode implantation (Fig. 1c). Meanwhile, in the control group (N = 7), animals underwent identical two-photon imaging sessions without LIPUS treatment. For LIPUS treatment, the transducer was positioned 14 mm from the cover glass with a polyvinyl alcohol (PVA) hydrogel cone attached (Fig. 1d).

Preceding the insertion of the microelectrode, microglia exhibited a uniform distribution, and no discernible morphological alterations were observed in the cortical tissue caused by the craniotomy. However, the inherent limitations of two-photon microscopy have historically posed challenges when studying the tissue response to electrodes inserted perpendicularly into the brain. These challenges have been previously addressed in published work[38]. Specifically, a probe was inserted into the cortex at an angle of 30° and secured with a chronic imaging window (Fig. 1d) to enable two-photon visualization of fluorescently labeled microglia. To minimize bleeding, the insertion process was carefully conducted to avoid large surface vasculature and

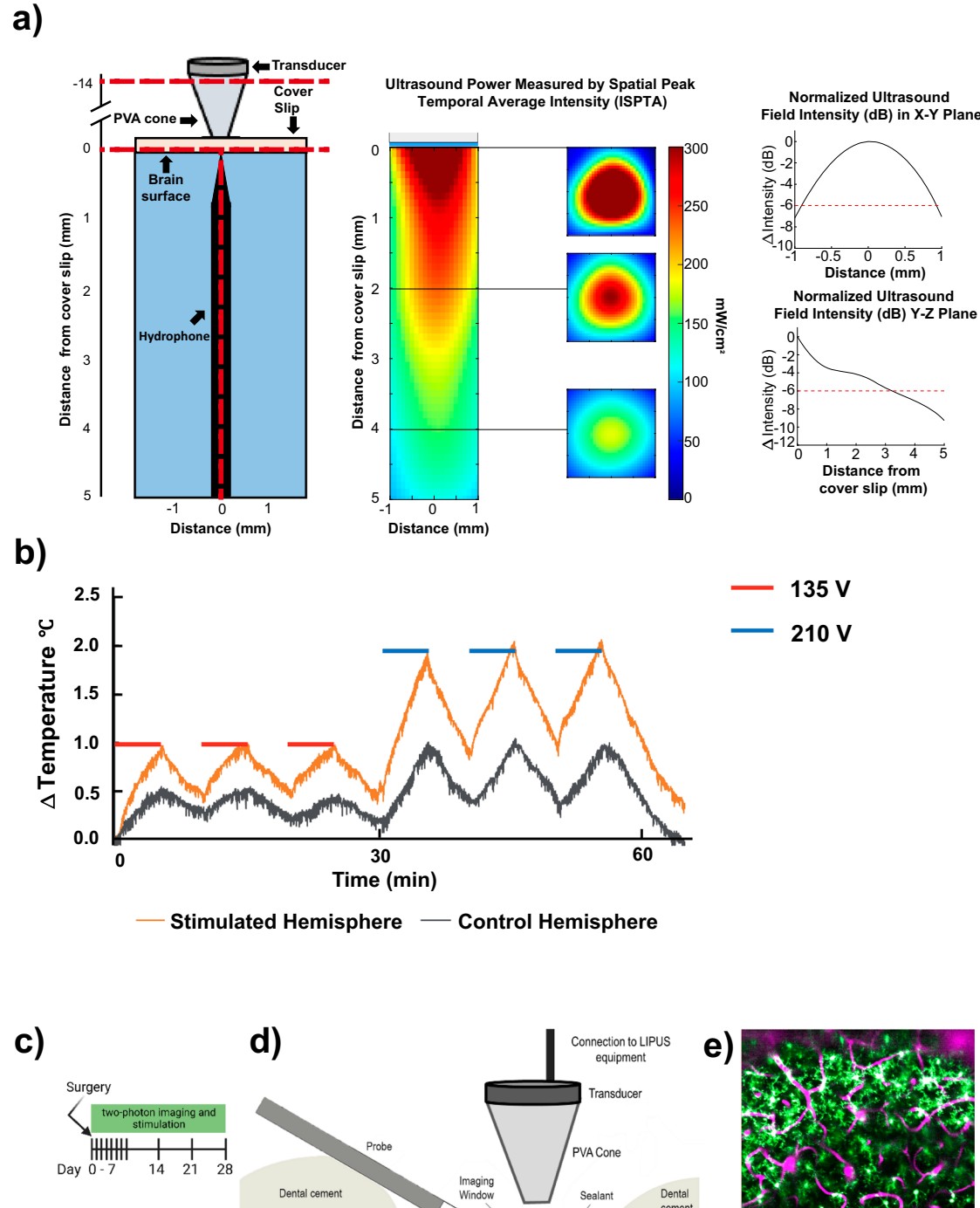

sealing techniques were employed to preserve the region of interest during chronic imaging sessions[27,37,84]. The laser power was kept at ~20 mW (never exceeding 40 mW) to prevent thermal damage. Photomultiplier tube (PMT) settings were adjusted accordingly for best image quality. For quantification purposes, a region of interest located 300 μm adjacent to the probe shank was selected (Fig. 1e). Furthermore, intraperitoneal administration of SR101 was performed to observe changes in the vasculature (magenta in Fig. 1e). This experimental protocol confirmed that LIPUS could be safely delivered to the brain and allow for the chronic imaging of microglia activity (green in Fig. 1e) using two-photon microscopy.

## LIPUS increased microglial migration velocity on day 1 and day 3

Microglial migration toward the injury site represents a key feature of the inflammatory response following microelectrode implantation[34,38,40]. Given the potential of LIPUS to attenuate the inflammatory response, our initial investigation focused on whether LIPUS could influence microglial migration. To explore this possibility, we aligned the images taken at different time points (see the "Methods" subsection "Mouse two-photon imaging"). Subsequently, we generated a composite image in which the image from the earlier time point was rendered in magenta, while the image from the later time point was displayed in green. This alignment allowed us to estimate the

**Fig. 1 | Experimental apparatus for LIPUS treatment and two-photon imaging of microglia and vasculature following microelectrode implantation. a** To evaluate LIPUS stimulation power, a submersible hydrophone preamplifier was placed in degassed water (blue area) below the coverslip and PVA cone. The measurement of the spatial-peak temporal-average intensity (ISPTA) was conducted by setting the transducer's voltage supply to 135 V. Axial and lateral ultrasound intensity plot from a representative trial ($N = 1$) used to characterize the transducer used for in vivo LIPUS experiments. Power was around 300 mW/cm² at the center of the plane, 1 mm below the cover glass. Imaging planes covered a depth range from 0 to 0.3 mm below the cover glass. Attenuation of ultrasound power was assessed in both the $X$–$Y$ plane (Top right figure) and the $Y$–$Z$ plane (bottom right figure). Degree of attenuation was quantified by comparing the power at the various locations to the power at the center of the $X$–$Y$ plane, specifically at the putative brain surface beneath the coverslip. **b** Representative trial ($N = 1$) of LIPUS induced tissue heating at two transducer operating intensities used for quantification of thermal effect from LIPUS stimulation. LIPUS was produced at 135 V (red) and 201 V (blue), a total of 15 min of LIPUS exposure, divided into three intervals of 5 min each, with 5 min of non-exposure between each sonication (red and blue periods). Driving the transducer at 135 V kept changes in brain temperature below 1 °C tested at 1.5 mm below the cortical surface. This driving voltage was used for subsequent experiments. **c** Illustration presents a timeline depicting the sequence of surgery, stimulation, and two-photon imaging. **d** Schematic representation of microelectrode implantation, sealing of craniotomy window, and LIPUS stimulation. Adapted from[38]. **e** A representative two-photon image of a Cx3CR1-GFP transgenic mouse following an I.P. injection of sulforhodamine 101 (SR101) showing microglia cells in green and cerebral blood vessels in magenta. Shaded blue region indicates the location of the implanted microelectrode. Scale bar = 100 μm.

microglial migration by measuring the displacement of each cell within the composite image during the interval between these two-time points. The analysis revealed that LIPUS treatment significantly increased the microglial migration velocity on day 1 ($1.35 \pm 0.07$ vs. $0.59 \pm 0.04$ μm/hr) and day 3 ($1.62 \pm 0.06$ vs. $0.96 \pm 0.05$ μm/h) but led to a reduction in velocity from day 4 to day 6 (Fig. 2b).

To visualize the spatial characteristics of microglia migration, the velocity of each individual microglia was plotted against the corresponding distance from the implantation site (Fig. 2c–f). Generally, there was a negative slope in the scatter plot, indicating that migration velocity tends to decrease as the distance from the implantation site increases. Notably, microglia close to the implantation site exhibited a higher velocity in the LIPUS group on day 1, as evidenced by a significantly higher intercept in the linear regression model (Fig. 2c). Our analysis demonstrated that LIPUS treatment significantly increased the velocity of microglia migration toward the implantation site during the early time points, but it resulted in a decrease in velocity at later stages (after day 4). These findings shed light on the directed movement of microglia toward the injury site and the benefits of LIPUS in boosting their migratory behavior.

## LIPUS reduced the morphological activation of microglia on day 6

After observing significant differences in microglial migration toward the injury site, we questioned whether LIPUS could also attenuate microglial morphological activation[38] resulting from microelectrode implantation. We classified the microglia into two stages: transitional stage (0) and ramified stage (1) based on previous studies[16,38]. Microglia were sampled within a range of 0–400 μm from the probe and fitted into a logistic regression model (Fig. 3a). $Y$-axis represents the percentage of ramified microglia at each distance bin (50: 0–50 μm, 100: 50–100 μm, 150: 100–150 μm, 200: 150–200 μm, 250: 200–250 μm, 300: 250–300 μm, 350: 300–350 μm, 400: 350–400 μm). Values closer to 1 indicate a higher proportion of ramified microglia, suggesting less morphological activation. Generally, ramification values increased with distance from the electrode surface in both LIPUS and control groups, confirming that morphological activation decreased with distance. To determine the suitability of fitting microglial ramification with a logistic regression model, the receiver operating characteristic (ROC) curve was plotted (Fig. 3a). ROC curves were created by plotting the true positive rate against false positive rate for various threshold settings used to classify observations as positive or negative (detailed calculation documented here[85]) and was applied to analyze the ramification plots. The diagonal random classifier line (black dashed line) indicates the performance expected from a classifier that makes predictions randomly, with no regard for the underlying data distribution. Closer distance between the ROC curve and random classifier line Indicates a less suitability for fitting ramification data with a logistic regression model. However, the ROC curve began to approach the random classifier line (black dashed line) by day 6 in the LIPUS group

and by day 7 in the control group (Fig. 3a), which could be due to the presence of more ramified microglia near the microelectrode starting day 6. Transition from a transitional state to a ramified state occurred earlier in the LIPUS group (on day 6) compared to the control group (on day 7), implying that LIPUS advanced the microglial transition from a transitional/activated state to a ramified state.

The degree of microglia activation was further evaluated using transitional (T-) and directionality (D-) indices (see the "Methods" subsection "Microglial activation and morphology"), based on the length or number, respectively, of leading (towards) versus lagging (away) microglia processes, as described in previous studies[34]. Indices closer to 1 indicate a ramified state as there are equal lengths or number of processes facing toward and away from the probe, while indices closer to 0 suggest a more activated state due to a greater length or number of processes preferentially oriented towards the implant. Occasionally, there are values greater than one, indicating a preferred orientation of processes away from the probe, potentially due to population variability or directed process extension toward a damaged blood vessel caused by tissue strain or inflammation[29,30,86]. Smaller values for T-index and D-index indicated a higher level of microglial activation. Similarly high levels of microglia activation were seen near the implantation site in both LIPUS and control groups. This observation was consistent with previous findings[34,37], indicating that the morphological activation of microglia was more pronounced in proximity to the implantation site (Fig. 3c, d). Notably, on day 6, the microglia processes exhibited significantly greater length and a higher T-index in the LIPUS group compared to the control group (Fig. 3b: within 50 μm, $1.04 \pm 0.04$ vs. $0.88 \pm 0.03$, $p < 0.05$; 50–100 μm, $1.02 \pm 0.02$ vs. $0.75 \pm 0.04$, $p < 0.001$). On the same day, the LIPUS group showed a significantly greater number of processes as indicated by a higher D-index compared to the control group (Fig. 3c: 50–100 μm, $0.95 \pm 0.01$ vs. $0.81 \pm 0.04$, $p < 0.05$). Therefore, our results revealed that LIPUS suppressed the morphological activation of microglia, especially on day 6 within 100 μm from the implantation site.

## LIPUS increased microglial expansion/retraction and total surveillance

After observing significant LIPUS-induced differences in microglial migration and morphology, we asked whether those changes affect microglial surveillance of the surrounding tissue areas. To answer this question, we evaluated the continuous surveillance activity of microglia[87]. Average surveillance area expansion (blue in top row)/retraction (magenta in top row) and total surveillance area (blue in bottom row) were analyzed (Fig. 4a) based on previously published methods[88]. Specifically, the average expansion/retraction represents the rate of surveillance, while the total surveillance reflects the area monitored by microglial processes. Expansion/retraction speed and the total surveillance area initially increased then started to decrease from day 2 (Fig. 4b). Providing evidence supporting LIPUS as

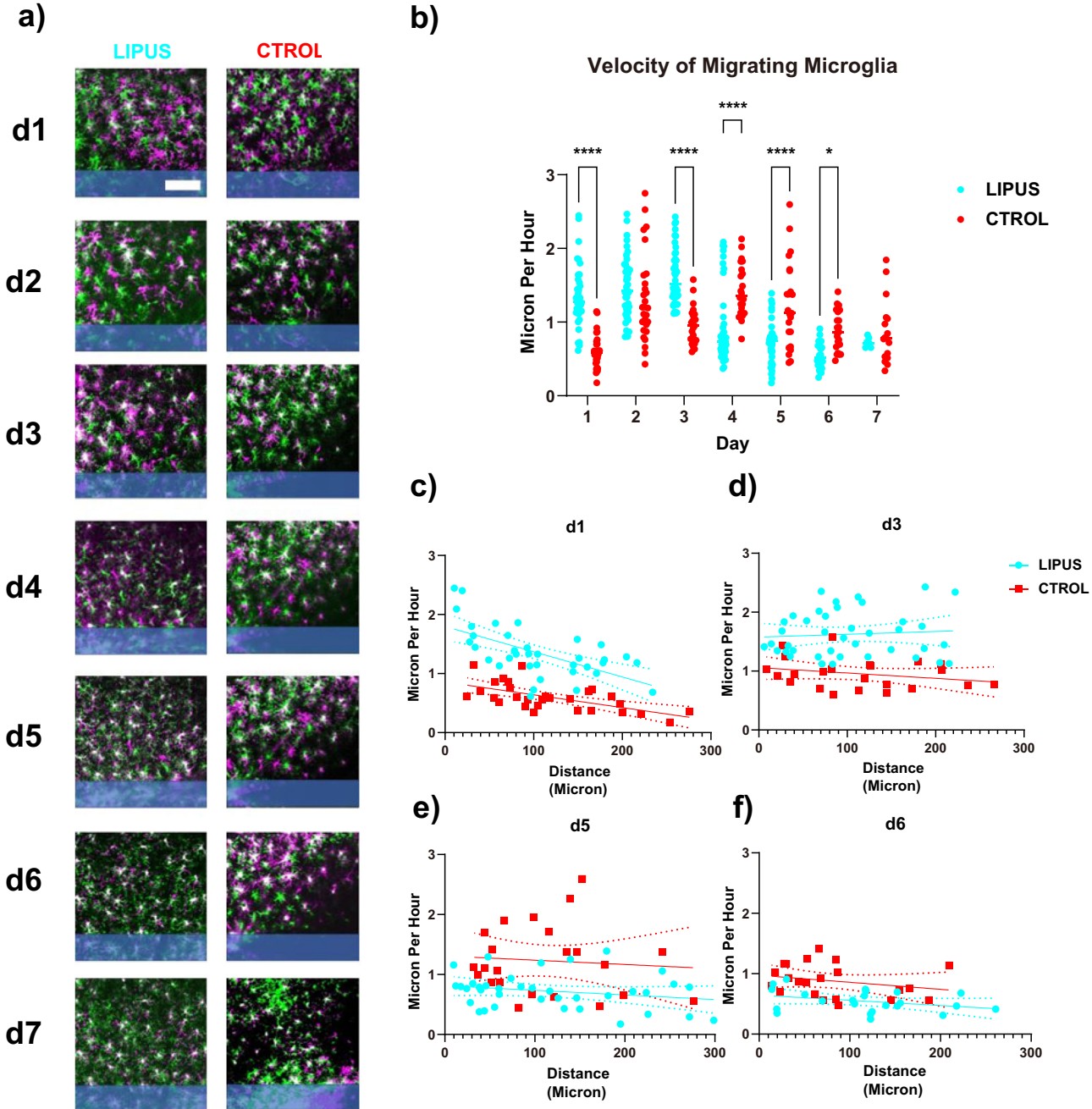

**Fig. 2 | LIPUS increased the velocity of migrating microglia on day 1 and day 3.** **a** Microglia migration was characterized by aligning images from an earlier time point (magenta) and the listed time point (green). White indicates no cell movements. After microelectrode implantation, microglia migrated toward the microelectrode (shaded blue) as indicated by green cells being closer to the probe compared to magenta cells. **b** The velocity of migrating microglia was quantified. LIPUS treatment significantly increased the velocity of migrating microglia on day 1 and day 3 but significantly decreased on days 4, 5, 6 (two-sided Šídák's multiple comparisons test, $*p = 0.0360$, $****p < 0.0001$). **c–f** spatial characterization of migrating microglia velocity on days 1, 3, 5, and 6. Each microglia was fitted to a linear regression model to examine the relationship between microglia migration velocity and the distance from the microelectrode (solid line represents best fit, and dotted lines represent 95% confidence interval bands). Significant difference was detected in the intercepts from the linear regression model ($p < 0.001$) on day 1. LIPUS $N = 7$, $n = 40$; Control $N = 5$, $n = 29$ (see Supplementary Table 1). Scale bar = 100 μm.

gliomodulatory, LIPUS significantly increased the expansion ($93.15 \pm 8.77$ vs. $44.50 \pm 6.86$ μm²/min), retraction ($101.84 \pm 7.58$ vs. $47.80 \pm 8.13$ μm²/min), and total surveillance area ($673.29 \pm 50.81$ vs. $286.43 \pm 46.21$ μm² per 10 min) compared to control group on day 7 ($p < 0.001$).

While we observed an increased surveillance activity of microglia in the LIPUS group, we want to determine whether this was due to the larger soma area of microglia in the LIPUS group. It was noted that

microglia near the implantation site were smaller compared to those away from the implantation site[89]. To account for this size effect, the average expansion/retraction speed and the total surveillance area were normalized by the size of the stable part of the microglia (Fig. 4c). A significant increase in the average expansion speed ($49.64 \pm 3.24$ vs. $32.40 \pm 3.50$%/min, $p < 0.05$), retraction speed ($51.01 \pm 2.82$ vs. $33.82 \pm 3.04$%/min, $p < 0.01$), and the total surveillance area ($338.17 \pm 21.05$ vs. $205.59 \pm 19.02$% per 10 min, $p < 0.001$) was still

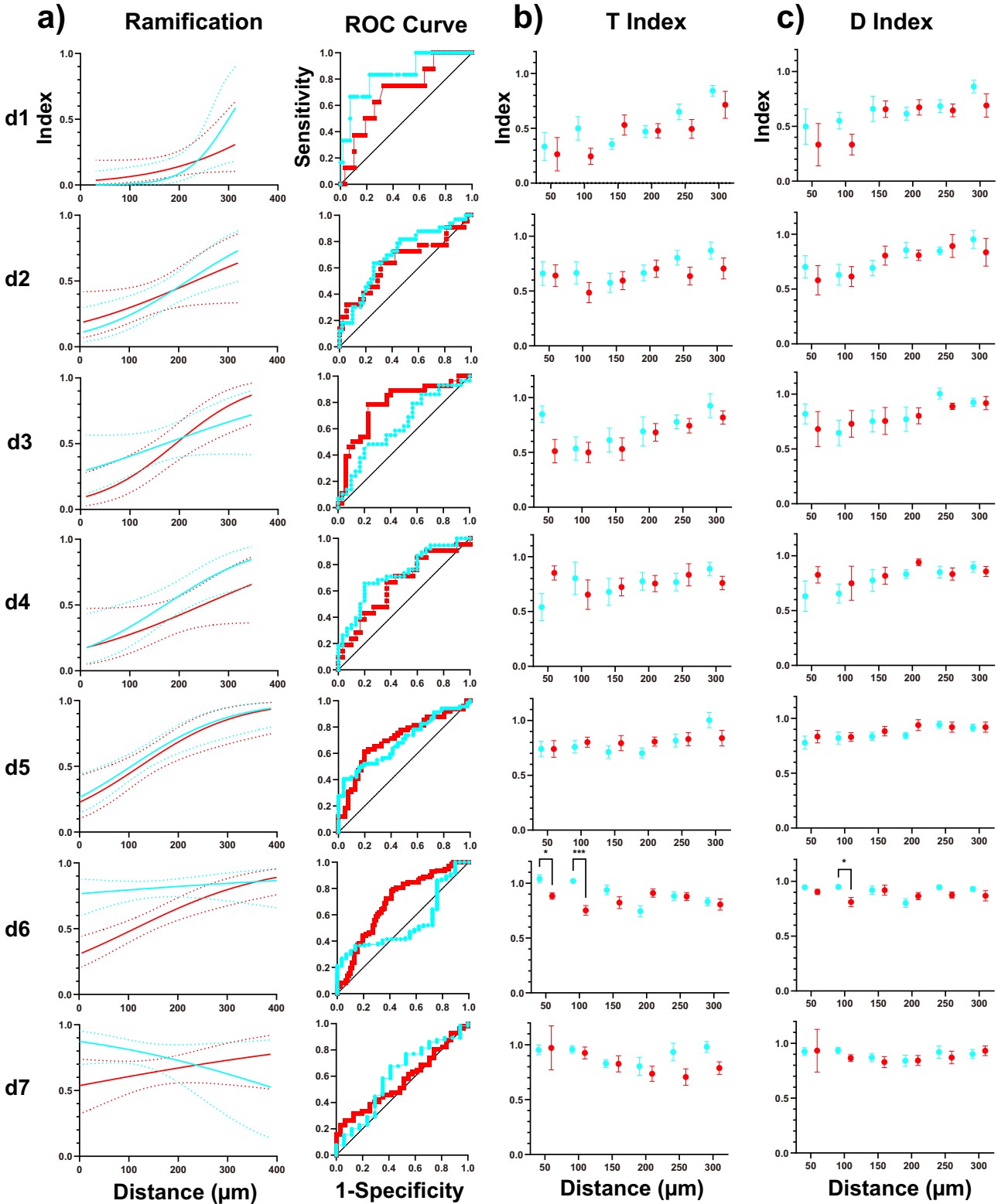

observed in the LIPUS group on day 7 (Fig. 4c). The spatial analysis of microglia expansion/retraction and surveillance indicated that LIPUS treatment increased the average expansion/retraction and total surveillance area on day 7 across all distances compared to the control group (Fig. 4d). Together, our findings demonstrate that LIPUS increased the microglia directed movements without affecting ramification or surveillance at earlier points (day 1, day 3), while promoting ramification and surveillance at later time points (day 6, day 7).

**LIPUS reduced the probe coverage from day 6**

We next calculated how LIPUS treatment affects microglial encapsulation of the microelectrodes using established techniques[34,36–38]. We quantified the percentage of probe coverage by calculating the ratio of the area occupied by GFP-positive microglia to the area of tissue directly above the probe shank (20 μm z-projection)[34,36]. A higher percentage of probe coverage indicates greater microglia encapsulation. We observed a rapid increase in the percentage of probe

**Fig. 3 | LIPUS attenuated microglial activation on day 6. a** Left: Logistic regression analysis of microglia ramification over distances for the LIPUS (blue) and Control (red) groups. *Y*-axis represents the predicted percentage of ramified microglia using the logistic regression model. Dotted lines represent the 95% confidence bands. Right: Receiver Operating Characteristic (ROC) curve for each logistic regression model used in left panels. The proximity to the black dashed line indicates the model's performance. Closer distance between the ROC curve and the random classifier line (black dashed line) indicates less suitability for fitting ramification data with a logistic regression model. Based on the ROC curves, the microglia ramification data was unsuitable for fitting into the logistic regression model on day 6 in the LIPUS group and on day 7 in the control group. **b** T-index

calculation, which assesses the degree of activation based on the length of the most prominent leading versus lagging microglia process relative to the electrode. Error bars indicate the standard errors. **c** D-index calculation, which measures the degree of activation based on the number of leading versus lagging microglia processes relative to the electrode. Across spatial bins, the T-index and D-index generally decreased with time until day 3, indicating a higher degree of microglia activation. Starting from day 5, both indices began to increase. Notably, LIPUS treatment significantly increased both the T-index and D-index on day 6 (two-sided Šídák's multiple comparisons test, $*p < 0.01$, $***p < 0.001$), providing clear evidence of attenuated microglial morphological activation. LIPUS $N = 5$, $n = 29$; Control $N = 4$, $n = 29$ (see Supplementary Table 2). Error bars indicate SEM.

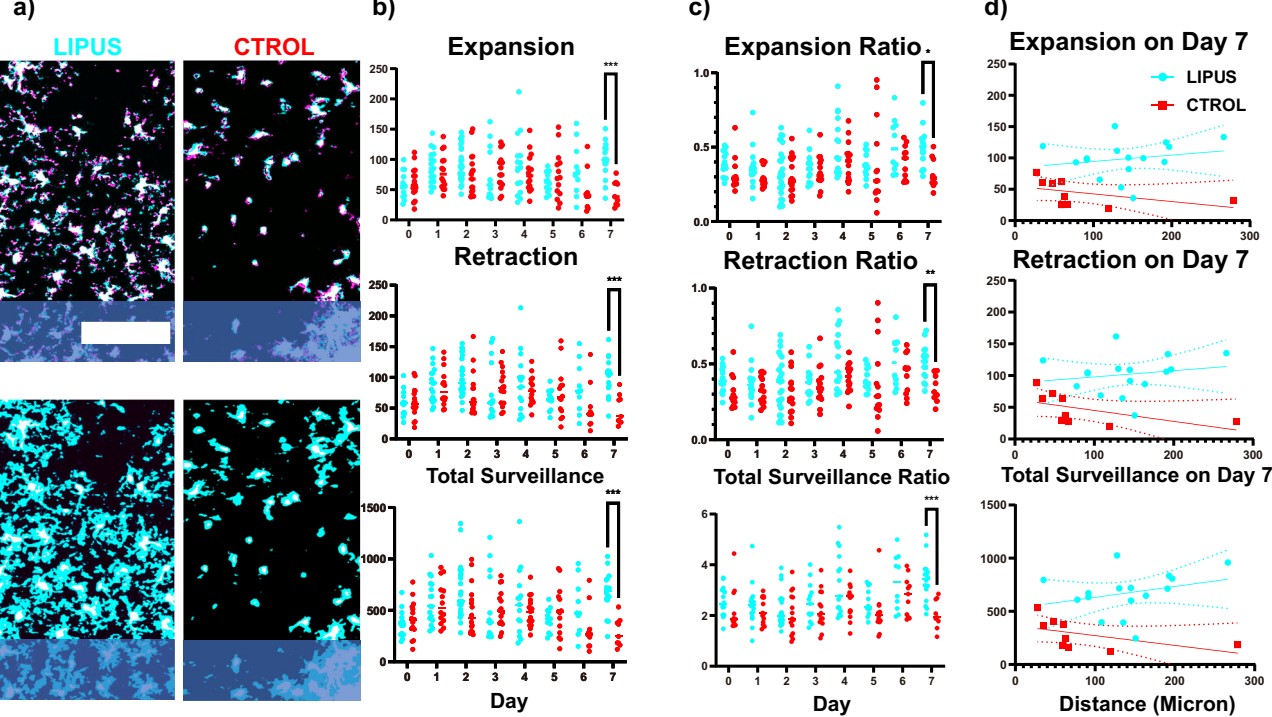

**Fig. 4 | LIPUS increased the surveillance area expansion/retraction and total surveillance area of microglia on day 7. a** Top two rows: Examples of two-photon processed images showing microglia expansion (cyan), retraction (magenta), stable part (white) over a 1-min interval. Bottom two rows: Examples of two-photon processed images displaying the total surveillance area (cyan) and stable part (white) of microglia over a 10-min period. **b** Quantification of expansion, retraction, and total surveillance area. LIPUS treatment significantly increased the expansion, retraction, and total surveillance area on day 7 (two-sided Šídák's multiple

comparisons test, $***p < 0.001$). **c** The expansion, retraction, and surveillance normalized by the stable part of microglia. LIPUS treatment significantly increased the expansion ratio, retraction ratio, and total surveillance ratio on day 7 (two-sided Šídák's multiple comparisons test, $*p = 0.01$, $**p = 0.004$, $***p < 0.001$). **d** Spatial characterization of microglia expansion/retraction and surveillance on day 7 (solid line represents best fit and dotted lines represent 95% confidence interval bands). LIPUS $N = 4$, $n = 16$; Control $N = 2$, $n = 9$ (see Supplementary Table 3). Scale bar = 100 μm.

coverage in both groups from day 0 to day 5. However, the LIPUS group exhibited faster stabilization and reduced encapsulation of the probe around day 6 (Fig. 5b, $22.34 \pm 3.05$ vs. $39.08 \pm 4.41\%$, $p < 0.01$), compared to the control group. By day 28, the LIPUS group had a significantly lower percentage of probe coverage compared to the control group ($15 \pm 5.80$ vs $51.68 \pm 2.58\%$, $p < 0.001$). Taken together, LIPUS treatment leads to faster stabilization (as early as day 6) and a lower percentage of probe coverage by microglia, indicating reduced microglial encapsulation of microelectrodes.

### LIPUS reduced the number of vessel-associated microglia on day 7

The implantation of microelectrodes not only induces microglia encapsulation but also leads to the leakage of BBB[16]. This increased BBB leakage is temporally correlated with enhanced microglia-vessel interactions[56]. To assess the impact of LIPUS on microglia–vessel

interactions, we quantified the total microglia population, vessel-associated microglia (defined as microglia extending at least one process onto vasculature), and their ratio over time post-implantation (Fig. 6). In both the LIPUS and the control group, the density of the total microglia and vessel-associated microglia decreased from day 0 to day 3, followed by an increase from day 3 to day 7 (Fig. 6b and c). Additionally, the vessel-associated microglia ratio peaked at 80% by day 2 and then gradually decreased to around 40% on day 6 for both the LIPUS and the control group (Fig. 6d). However, while the LIPUS group maintained a vessel-associated microglia ratio of approximately 40% on day 7, the median ratio in the control group rose above 70% (Fig. 6d, $40.43 \pm 3.87$ vs. $70.67 \pm 6.15\%$, $p < 0.0002$). To account for individual differences, we analyzed the ratio change by subtracting the ratio on day 0. LIPUS treatment led to a notable and significant decrease in the ratio change of vessel-associated microglia on day 7 (Fig. 6e, $-32.62 \pm 6.18$ vs. $3.72 \pm 4.07\%$, $p < 0.0001$). These findings

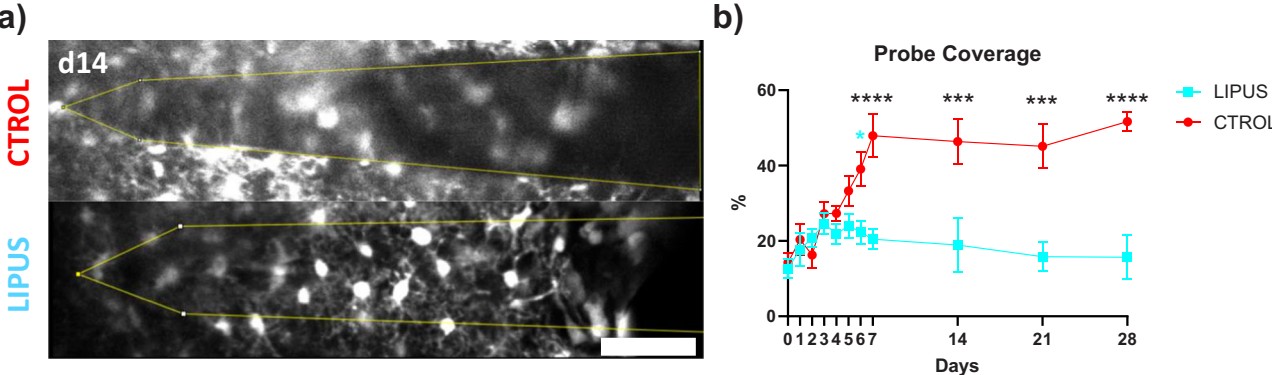

**Fig. 5 | LIPUS attenuated microglial encapsulation of intracortical microelectrode starting day 6. a** Representative two-photon images on day 14 revealed that microglia extensively covered the probe in the control group, whereas in the LIPUS group, microglia displayed distinct processes with reduced overlap. **b** The percentage of microglial surface coverage was quantified up to 20 μm above the surface of the implant (yellow outline in a), comparing LIPUS and control groups.

LIPUS treatment resulted in a lower level of probe coverage, which occurred around day 6, and remained consistently lower than that in the control group (two-sided Šídák's multiple comparisons test, *p = 0.01865, ***p < 0.001, ****p < 0.0001). LIPUS N = 7; Control N = 6 (see Supplementary Table 4). Error bars indicate SEM. Scale bar = 50 μm.

suggest that LIPUS may mitigate the accumulation of microglia on nearby blood vessels during the inflammatory phase.

**LIPUS reduced the diameter of cerebral blood vessels on day 28**
While the role of microglia in the regulation of vasculature has only recently begun to be elucidated[56,90], it is evident that microglia does interact with vasculature[90]. We have observed in our previous studies that cerebral blood vessels dilate following microelectrode implantation[27,39,40,51,56,90], which was temporally correlated with the activation of microglia. Hence, we tested whether the reduction in microglial activation and vessel-associated microglia ratio by LIPUS might contribute to the prevention of blood vessel dilation chronically. The diameters of blood vessels near the implantation site were evaluated using VasoMetrics ImageJ plugin[91] (Fig. 7). We found a steady increase in the average blood vessel diameter during the first 6 days. From day 7 to day 28, the control group exhibited a slight increase, while the LIPUS group showed a slight decrease (Fig. 7b). LIPUS significantly reduced the vessel diameter on day 28 (Fig. 7b, 3.95 ± 0.13 vs. 4.76 ± 0.14 μm, p < 0.05). To eliminate the possibility that the reduction in blood vessel diameter was a result of the initial vessel selection, we carefully selected blood vessels with similar diameters on day 0 in both groups (Fig. 7c). LIPUS significantly reduced vessel diameter only after LIPUS stimulation (Fig. 7c, 4.37 ± 0.17 vs. 4.54 ± 0.16 μm, p < 0.05). We tested whether LIPUS also affected the density and structure of the vessels. We did not detect a significant difference in vessel area coverage (Fig. 7d), indicating the density of vessels was unaffected by LIPUS. We further quantified tortuosity vessel branch length, and number using the skeletonized macro in ImageJ[91,92] and found that LIPUS did not significantly alter vessel structure (Fig. 7e–h). Taken together, our findings suggest that LIPUS treatment reduces vessel dilation but does not affect the structure of the cerebral blood vessels.

**LIPUS improved neuronal SU activity and decreased astrocytic scarring**
To examine the influence of LIPUS on the functionality of the implanted microelectrode, electrophysiological data was recorded from L5 of lightly anesthetized rats through somatosensory stimuli. We examined the single-unit (SU) recording between the LIPUS and Control groups independent of laminar depth and averaged all the channels. SU yield was calculated as a percentage of channels along the shank that detected at least one SU. SU Yield (32.81 ± 16.01%) in the LIPUS group was significantly higher than in the control (10.94 ± 10.94%) at the end of the 6-week study (Fig. 8a). Both groups

experienced a rapid decline in SU Yield in the first two post-operative weeks. LIPUS-treated group exhibited a stabilization at around day 29 while the control continued to exhibit a decline, which can also be observed in the number of active channels (Fig. 8b). Similarly, the SU signal-to-noise ratio (SNR), a measurement of the strength of SU activity, showed significant improvements between day 14–21 and at day 43 (Fig. 8c). Also, SU amplitude was consistently higher in the LIPUS group and significantly greater between 5 and 6 weeks (Fig. 8d). Average SU SNR in the LIPUS group was 6.56 ± 3.32 on day 43 compared to control at 0.533 ± 0.533. The average amplitude in the LIPUS group on the day at the end of 6 weeks was 155.31 ± 41.33 μV compared to 10.69 ± 10.68 μV for control. Noise floor remained relatively consistent in both groups, with the LIPUS group having a slightly higher noise profile during the first week (Fig. 8e, Day 0: 9.75 ± 0.95 vs. 5.85 ± 1.11 μV). There was no significant difference in device impedance between the LIPUS and control groups during the 6-week period (Fig. 8f, p = 0.2936). Together, these results suggest that LIPUS may increase the recording performance of L5 neural activity at later time points.

To determine whether LIPUS-modulated astrogliosis, immunohistochemical analysis was performed 6 weeks after implantation. Astrogliosis was examined by glial fibrillary acidic protein (GFAP) labeling. LIPUS significantly reduced intensity compared to the control group within the 50 μm from the probe (Fig. 8g, 1.64 ± 0.081 vs. 2.59 ± 0.14 μm). This lower activation of astrocyte cells around the probe further validates the lessening glial scarring and could explain some of the improvements seen in the electrophysiological data. Because brain-derived neurotrophic factor (BDNF) has been implicated as a potential mechanism related to LIPUS, we examined BDNF activity around the implant. The fluorescent intensity of BDNF showed no difference in the two groups (Fig. 8h). LIPUS showed increased recording performance in L5 neurons as well as reduced astrocyte FBR around chronically implanted arrays.

## Discussion
LIPUS has been shown to be a promising tool for suppressing the FBR[73] and microglia activation[93]. Our findings indicated that LIPUS treatment resulted in increased microglia migration on day 1 and day 3 (Fig. 2), more ramified morphology on day 6 (Fig. 3), enhanced microglia surveillance on day 7 (Fig. 4), decreased probe coverage from day 7 (Fig. 5), reduced vessel-associated microglia ratios on day 7 (Fig. 6), attenuated dilation of blood vessels on day

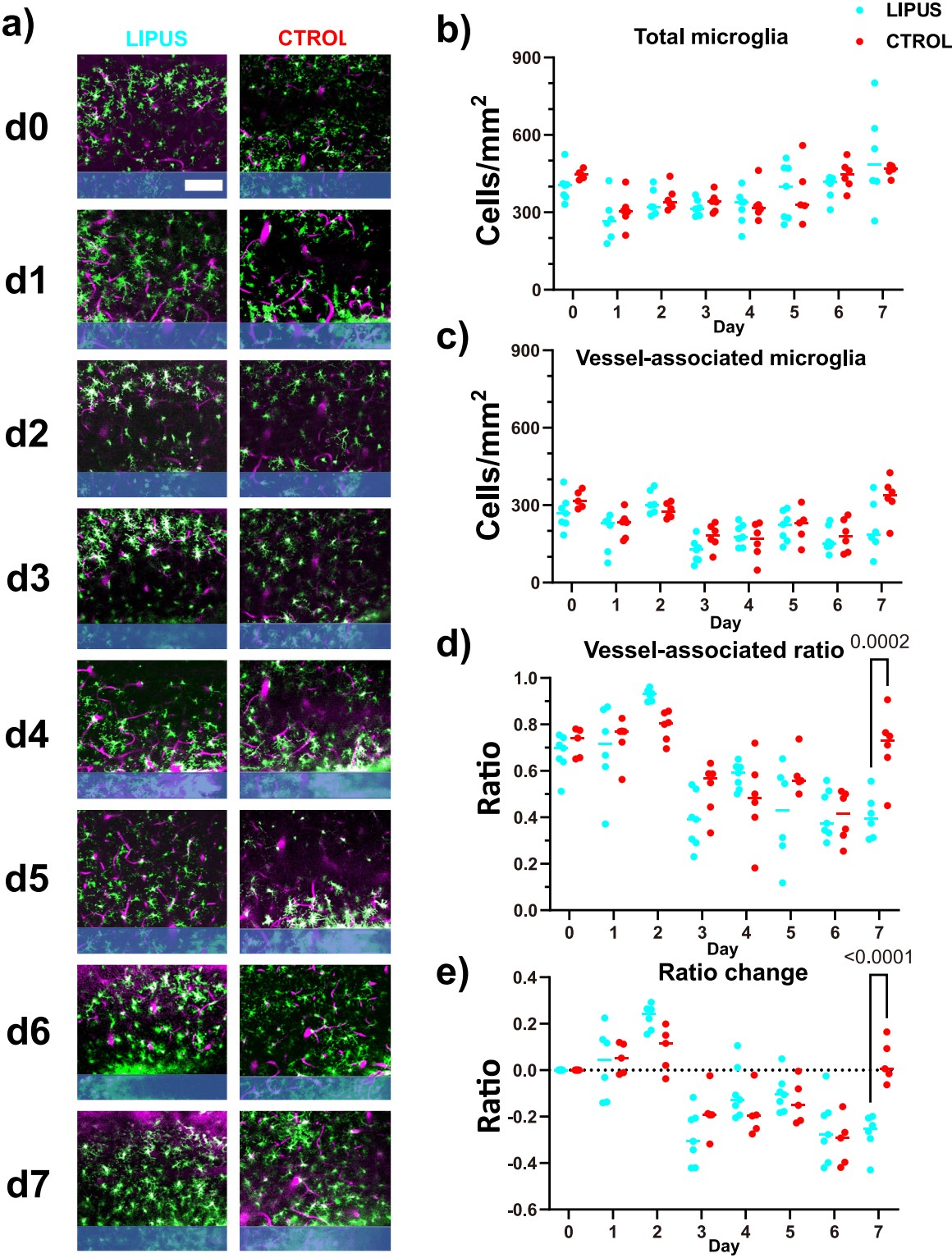

28 (Fig. 7), reduced astrocyte activation and scarring around the probe (Fig. 8), and improved chronic recording performance (Fig. 8).

The impact of LIPUS on improving long-term electrophysiological recordings was investigated 6 weeks after the implantation of a 4-shank intracortical microelectrode. Previous studies have indicated that unit recordings stabilize around this timeframe, which could be due to the late-onset phase of the FBR[94,95]. Our results exhibited a downward trend of the SU yield in both groups but around week

2 starts to stabilize and maintain consistently until the 6-week end-point which may be due to the decrease in astrocytic encapsulation of the shank (Fig. 8). Additionally, the SU SNR and amplitude were consistently higher in the LIPUS group compared to the control. This promising outcome underscores the potential of LIPUS as a therapeutic approach to address challenges associated with FBR in neural implantation. Within the CNS, microglia-mediated neuroinflammation has been hypothesized to be the main contributing factor in the FBR impeding tissue–microelectrode integration[38].

**Fig. 6 | LIPUS reduced the number of vessel-associated microglia on day 7.** To identify vessel-associated microglia, a 22-mm Z stack was analyzed to identify microglial processes that exhibited at least one attachment to the blood vessel. **a** Representative two-photon images showing microglia (green) and vasculature (magenta) from day 0 to day 7. **b** Total microglia density (including non-vessel-associated and vessel-associated microglia) initially decreased up to day 3 and then returned to the same level as day 0 in both LIPUS and control groups. **c** Vessel-associated microglia density also exhibited a similar trend, decreasing up to day 3 and then returning to the baseline level. **d** Vessel-associated ratio was calculated by dividing the number of vessel-associated microglia by the total number of microglia. LIPUS treatment appeared to increase the vessel-associated microglia ratio on day 2 and significantly decrease it on day 7 (two-sided Šídák's multiple comparisons test, $p = 0.0002$). **e** To account for individual variability, the ratio change was calculated by subtracting the vessel-associated ratio on each day from the vessel-associated ratio on day 0 for each animal. LIPUS treatment significantly reduced the vessel-associated microglia ratio on day 7 (two-sided Šídák's multiple comparisons test, $p < 0.0001$). LIPUS $N = 6$; Control $N = 6$ (see Supplementary Table 5). Scale bar = 100 μm.

During microelectrode implantation, the BBB is compromised, resulting in infiltration of blood contents which causes inflammation to neural tissue[51]. Molecules, such as fibrin[96] and ATP[97], released from the damaged area attract microglia, prompting them to migrate toward the injury site[51]. Previous studies showed that microglia initiate their migration within 12–24 h post implantation[36,40,45,46]. Our results show that microglia migration speed is higher in the LIPUS group compared to the control group on day 1 and day 3 (Fig. 2). Faster migration of microglia toward the implantation site suggests that LIPUS could promote microglia to facilitate the closure of the wounds caused by microelectrode implantation, thereby potentially mitigating further neuronal damage by reducing the infiltration of blood contents[98]. LIPUS may amplify the release of ATP[99], possibly originating from the injury site[100], connexin 43 hemichannels in astrocytes[101], and Pannexin 1 channels in vasculature[90]. ATP, in turn, activates P2Y12 receptors on microglia, thereby increasing the speed of their directed movements toward the damaged area[102].

Despite the increased migratory speed of microglia, our results did not show a change in microglia density between the two groups over time, as observations were limited to a 400 μm area, and therefore, changes in bulk density were not tracked similarly to previous two-photon studies[38]. Similarly, microglia forming lamellipodia sheaths were not counted due to difficulty in distinguishing individual microglia cells[38,84]. It is likely that microglia density near the implant is maintained as microglia adhere to the implant. We observed that microglia in the LIPUS group exhibited longer process lengths and a higher number of processes on day 6 (Fig. 4). Mechanistically, LIPUS may facilitate this transition by potentially breaking down fibrin polymers[77], thereby reducing the burden of clearance, or by enhancing glycolytic metabolism[103] through the activation of mechanosensitive channels[104]. Furthermore, our results indicate that enhanced microglia repair via LIPUS at acute time points (day 1 and day 3) (Fig. 2) limit the propagation of injury, thereby reducing microglia activation at chronic time points (day 6) (Fig. 3). The decrease in microglia activation at chronic stages may reduce the release of proinflammatory cytokines[74] and NO[105], while also mitigating harmful events such as lysosomal dysfunction[41,106], peroxidated lipid accumulation[107], and ferroptosis[108]. These proinflammatory processes may coordinate with over proliferation of microglia, leading to microgliosis and increased microglial probe coverage as the chronic neuroinflammation progresses[109–111].

If the FBR and inflammation are not resolved timely, events such as oxidative stress[112,113] and microglia priming[114] can lead to persistent activation of microglia. As a result, activated microglia secrete proinflammatory cytokines that lead to the formation of a glial scar[53]. The glial scar not only displaces neurons from the recording range[53,115] but also forms a physical barrier, inhibiting ion diffusion and preventing the propagation and detection of neural signals[18,51,84,116,117]. This encapsulation sheath, a hallmark of the foreign body response, is made up of microglia cells, astrocytes, fibroblasts, and extracellular build-up[53,118]. We observed reduced microglial probe coverage at later stages (starting day 6) in the LIPUS group (Fig. 5), indicating better interfacing between the tissue and implanted microelectrode[36]. LIPUS may be able to modulate astrocytes as we observed a decrease in GFAP fluorescence within 50 um of the shank (Fig. 8). This could be due to changes in microglia activation, as microglia cells are able to activate and recruit astrocytes via TNFα, IL1α/β, C1Q secretion and the down-regulation of P2Y1 receptors[119–121]. Astrocyte activity during the foreign body response is also modulated by different factors, including interleukins, TGF-β, NFκB, STAT3, and fibrinogen[122].

The FBR involves several signaling pathways that can be regulated by mechanosensitive and thermosensitive channels[123,124]. Ultrasonic waves delivered by LIPUS can open these channels, thereby mediating neuroinflammation. Whether the effects of LIPUS could be beneficial or detrimental to the resolution of the FBR largely depends on the underlying mechanisms and pathways it engages. For example, the activation of TRPA1 receptors may exacerbate myelin damage and reduce cognitive outcomes[125], while activating the TRPV1 receptor facilitates myelin repair, which is an important mechanism for functional recovery[126]. Several channels and receptors can modulate microglial activity including microglial migration, ramification, and surveillance[87,102,126,127]. Notably, the activation of TRPV1 receptors boosts the microglia migration[126], potentially contributing to the observed increase in microglial migration on day 1 and day 3 (Fig. 2). Similar to TRP channels, the activation of mechanosensitive channels such as Piezo1[127] can also result in enhanced microglial migration (Fig. 2) via cytoskeleton remodeling[128–132]. Future studies should investigate these mechanisms in closer detail as well as comparing the performance of LIPUS to other well-established intervention strategies, such as dexamethasone[37,133,134].

Previous studies have highlighted that FBR dilates local blood vessels[39,40], potentially due to the release of NO[105,135]. This abnormal vasodilation has been linked to exacerbated brain swelling[136], indicating a disruption in neurovascular coupling. LIPUS has been demonstrated to promote vascular remodeling and increase the vessel diameter and length on day 3 following distal middle cerebral artery occlusion[75]. Furthermore, astrocytes can facilitate further breakdown of the BBB through increased expression of connexon-43 triggering vascular leakage through VEGFR2 signaling[137,138]. Even though the vessel length was consistently higher in our LIPUS group from day 2 to day 6 (Fig. 7), we did not detect a significant difference between groups on each individual day (with the lowest $p$-value on day 4: $p = 0.24$), neither in vessel length nor in vessel diameter. Chronic reduction in vessel diameter on day 28, rather than an acute change (Fig. 7), implies that the benefits of LIPUS treatment are less likely to result from vascular remodeling. The observed decline in vasodilation at a chronic time point (Fig. 7) may be a result of the accelerated microglia repair (Fig. 2), reduction in persistent microglia and astrocytic activation (Figs. 4 and 8), and reduction of NO release over time[59], preventing the dilation of blood vessels. Future research should investigate the role of LIPUS on microglia–BBB interactions.

For example, microelectrode implantation leads to the infiltration of blood contents such as fibrinogen, due to the rupture of BBB[51]. The immunoreactivity of fibrinogen near the microelectrode was significantly higher than the distal region around 1 week[139]. Previous studies have shown that fibrinogen can trigger microglial and astrocytic activation[140,141], leading to the generation of reactive oxidative stress[96]. Elevated oxidative stress can lead to impaired microglial and astrocytic clearance ability[103]. While the over-proliferation of microglia may transiently compensate for the impaired clearance capabilities, it ultimately contributes to microglial senescence[142]. These underlying

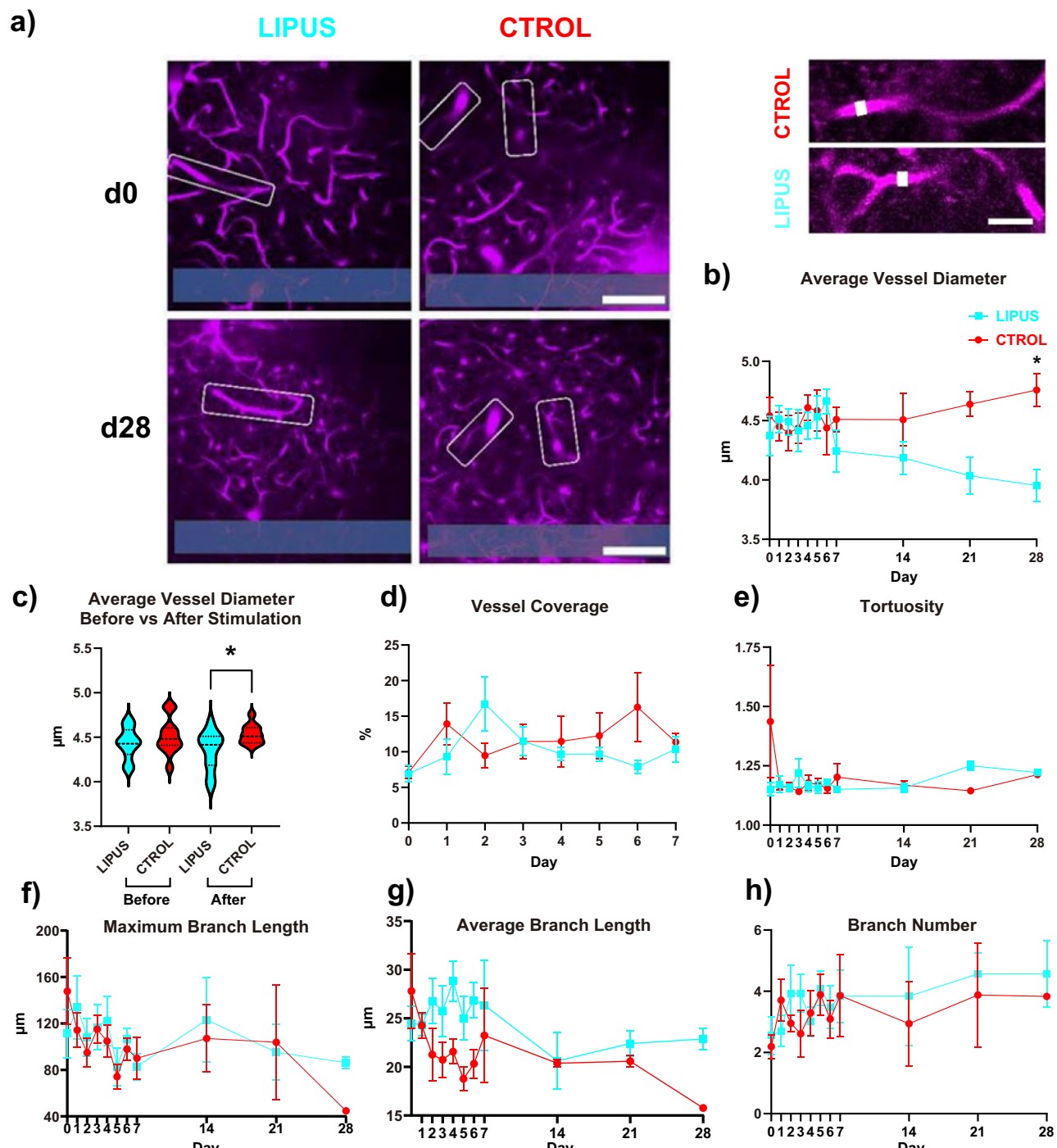

**Fig. 7 | LIPUS reduces the diameter of cerebral blood vessels on day 28. a** Left images: two-photon microscopy images of LIPUS and control group on day 0 and day 28. White boxes denote the analyzed vasculatures. The probe is outlined in blue at the bottom of the images. Scale bar = 100 µm. Right images: magnification of blood vessels on day 28 in both groups. Scale bar = 20 µm. **b** Average blood vessel diameter over 28 days. LIPUS treatment significantly decreases the average vessel diameter on day 28 (one-sided Tukey's multiple comparison test, *p = 0.02). **c** Average vessel diameter before and after stimulation on day 0. Significant reduction in average blood vessel diameter in the LIPUS group was observed only after stimulation compared to the control group (one-sided Welch's t-test, *p = 0.0151). **d** No significant difference between groups was detected in vessel area coverage percentage. **e** Tortuosity measures the level of twisting or distortion of the vessels. No significant difference between groups was detected in tortuosity. **f** Maximum blood vessel branch length showed no significant difference between groups. **g** A higher average branch length was observed in the LIPUS group compared to a control group from day 2 to day 7 and day 28, however, no significant difference was detected. **h** No significant difference between groups was detected in number of blood vessel branches per vessel. LIPUS N = 3, n = 7; Control N = 4, n = 5 (see Supplementary Table 6). Error bars indicate the SEM.

processes could explain the accumulation of activated microglia near the microelectrode, particularly at early stages (before day 6; Fig. 4). Recent studies have shown that LIPUS, in a similar manner to activated microglia, can drive VEGF expression, a vascular permeability factor that supports vascular formation[78]. Then as blood vessels are being repaired and fibrin and cell debris are cleared, microglia near the microelectrode start to transition back to a ramified state suggesting a resolution of inflammation (Fig. 4). Blood-derived macrophages and

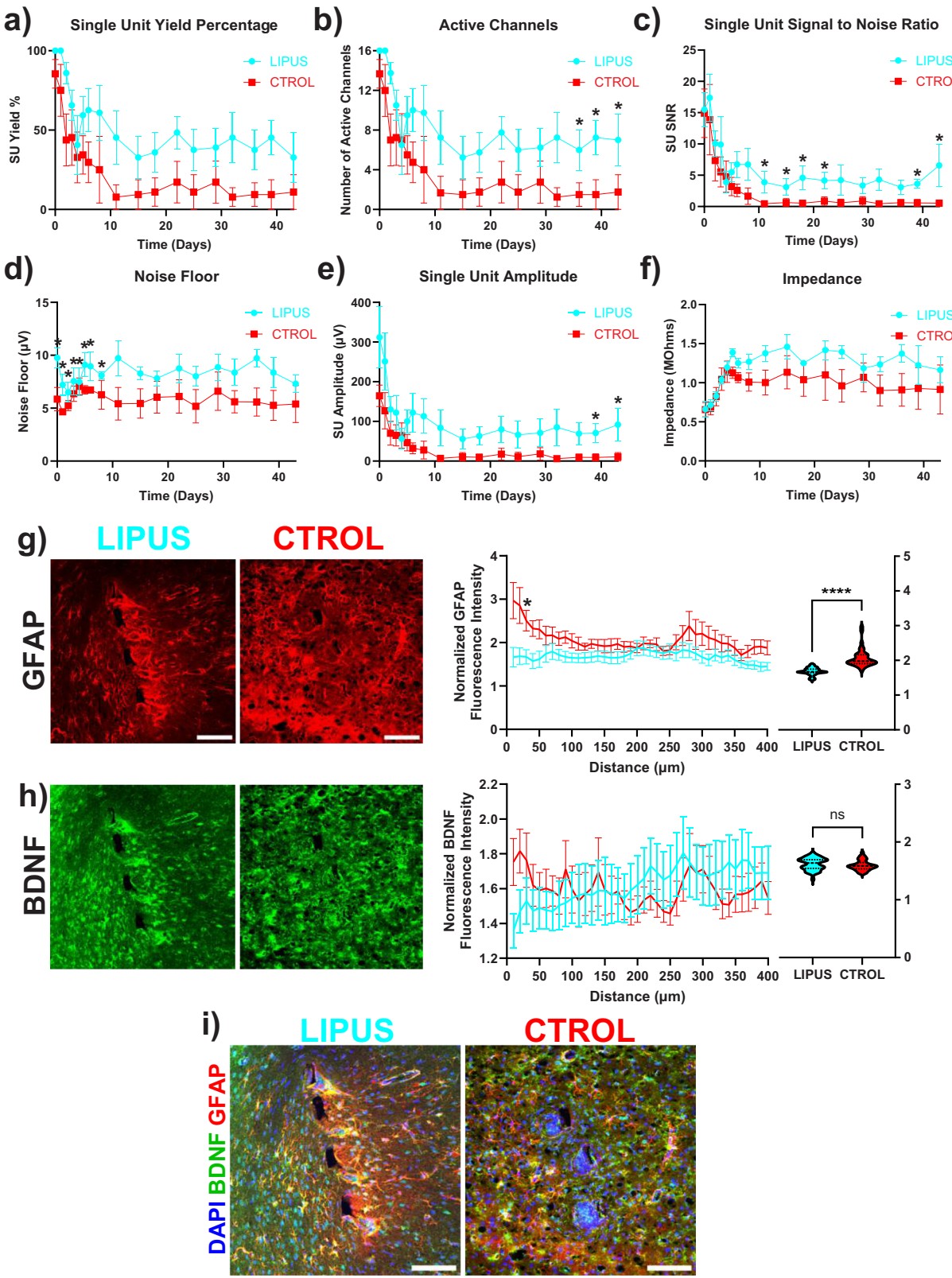

neutrophils also aid in the resolution of inflammation as they can clear pathogens and increase the permeability of the BBB[143,144]. Future studies would investigate more in-depth how blood vessel permeability and macrophages are affected by LIPUS treatment.

A few limitations exist within the study due to the restrictions of the experimental design and inherent data collected. First, a major limitation is the lack of detailed histological analysis at multiple depths

and a wider breadth of staining. Future studies should examine fibroblasts, neurons, oligodendrocytes, astrocytes, and other cells involved in the FBR in greater detail, in doing so would determine cellular and molecular pathways that LIPUS may be modulating. Furthermore, more extensive histology would allow for the observation of microglia proliferation as we were unable to do so within this study. Although we performed staining for BDNF and GFAP, they were only performed at a

**Fig. 8 | LIPUS improves neuronal single-unit activity and decreases astrocytic scar formation around the probe. a** Single-unit yield over time. **b** Number of active channels over time. **c** Single-unit SNR. **d** Single-unit amplitude. **e** Noise floor. **f** Average impedance. * Indicates significant group-wise differences via a linear mixed model likelihood ratio test with a 95% confidence interval (two-sided, $p < 0.05$). **g** Left image: Representative histological stain for GFAP around a microelectrode probe hole in LIPUS-treated and control animals; Middle: GFAP intensity is increased in the control group compared to the LIPUS-treated group 30 μm away from the probe; Right: violin plot quantifies GFAP within the first 50 μm

away from the shank (two-sided Sidak's multiple comparisons, ^$p < 0.0001$, *$p = 0.0371$; Welch's $t$-test ****$p < 0.0001$). **h** Left image: Representative histological stain for BDNF around a microelectrode probe hole in LIPUS-treated and control animals; Middle: BDNF staining showed no difference between the LIPUS and Control groups; Right: violin plot quantifies BDNF within 50 μm away from the shank. **i** Representative histological stain for DAPI, BDFN, and GFAP. **a**–**f**: LIPUS $N = 4$; Control $N = 4$. **g**, **h**: LIPUS $N = 3$, $n = 10$; Control $N = 3$, $n = 9$ (see Supplementary Table 7). Scale bar = 100 μm. Error bars indicate the SEM.

single depth (approximately cortical L5). A future study will need to be conducted to further characterize the attenuation of LIPUS over distance, as many neurological diseases and disorders not only affect the cortex but also deep brain structures.

An important future direction is how to translate LIPUS for BCI. An examination of how LIPUS would penetrate the tissue through the skulls of non-human primates and humans would be beneficial as the results will be widely different than in rodents. Additionally, examining whether LIPUS will work together with deep brain stimulation or single element and array transducers, will expand the usage of LIPUS within the clinical space. For deeper implants, focused ultrasound or LIPUS through an array of ultrasound transducers could be used to focus ultrasound waves to deep brain targets, such as for deep brain stimulation electrodes, without overstimulating shallower brain regions[145].

To optimize the application of ultrasound stimulation for both the bio-integration of BCIs and the treatment of neurodegenerative diseases, future studies should explore the parameter space of ultrasound stimulation to achieve optimal therapeutic benefits. These parameters include intensity[146], frequency[147], pulse repetition frequency[148], and duty cycle[149]. Among these parameters, it was observed that higher-intensity ultrasound led to an increase in blood flow[150]. However, excessively high intensity could potentially breach the BBB[151]. Furthermore, different frequencies of ultrasound have the potential to induce distinct biological responses. For example, lower frequencies (e.g. 350 kHz) activate TRPA1 channels, while higher frequencies (43 MHz) stimulate piezo1 channels[147]. Moreover, the adjustment of pulse repetition has the potential to selectively modulate excitatory and inhibitory neurons[148], providing a promising approach to precise neuromodulation. In addition to parameter considerations, the variations in bone density within the skull pose challenges for precise targeting of LIPUS to the intended treatment area[152]. Future studies should focus on refining the models of ultrasound propagation through the skull[153] and developing phase arrays[154] to steer focal ultrasound to improve the precision of LIPUS treatment.

This study investigated the potential of low-intensity pulsed ultrasound (LIPUS) as a therapeutic intervention for mitigating the foreign body response induced by the implantation of intracortical microelectrodes. We found that LIPUS treatment effectively enhanced microglial migration toward the implantation site. While microglial ramifications and surveillance remained initially unchanged, LIPUS treatment facilitated the transition from active to surveillance states within one week of daily treatments. As a result, LIPUS significantly reduced the microglial encapsulation of the microelectrode, lowered the vessel-associated microglia ratio, and reduced the average vessel diameter during the chronic stage. Lastly, LIPUS reduced astrocytic scarring around the probe and increased recording performance chronically. These findings suggest that LIPUS treatment has the potential to attenuate glial encapsulation of intracortical microelectrodes.

## Methods
### LIPUS treatment
**Equipment to generate LIPUS.** LIPUS was generated by a function generator (Keysight, Santa Rosa, CA, USA), a power amplifier (RF

Power Amplifier 2200L, E&I, Rochester, NY, USA), and a single-element focused transducer (#1219, 851 Material, Wrap Electrode, American Piezo, Mackeyville, PA). The transducer was driven at its first longitudinal resonance frequency, 1.13 MHz, using a 135 V peak-to-peak sine wave, with a pulse repetition frequency (PRF) of 2 Hz at a 4.4% duty cycle (22 ms burst duration). Hydrophone (HNC 1000, Onda Corporation, Sunnyvale, CA) in degassed water was used to measure the spatial peak temporal average intensity ($I_{SPTA}$) of the transducer through a glass coverslip used during imaging ~300 mW/cm$^2$ (Fig. 1a) to avoid tissue heating (Fig. 1b). To direct the acoustic beam to the targeted region (visual cortex) of the brain, the transducer was positioned using a stereotaxic apparatus.

For the rat cohort, immediately following electrode implantation, subjects receiving LIPUS treatment had a piezoceramic transducer with freshly made PVA acoustic horn coupled to dental cement using ultrasound transmission gel (Aquasonic 100, Parker Laboratories).

**Acoustic coupling medium for LIPUS stimulation.** In order to prevent air from interrupting the wave path, a polyvinyl alcohol (PVA) cryogel was applied as an acoustic coupling medium for LIPUS[63]. To prepare the acoustic coupling medium, the PVA powder (PVA: U228-08, Avantor, Center Valley, PA) was subjected to a passive melting process in degassed water at a concentration of 1 g of PVA per 10 mL of water. Melting was achieved by placing the PVA powder in a capped container to minimize evaporation. Subsequently, the container was immersed in a water bath to control the temperature below 100 °C. The resulting solution, consisting of PVA and water, was then carefully dispensed into cone-shaped molds with geometry optimized for focused delivery of LIPUS. The solution solidifies and strengthens through two consecutive freeze-thaw cycles. Each cycle consisted of freezing the molds at −20°C for 10 h and then thawing at room temperature for 14 h. Finally, the PVA cones formed from the molds were stored in degassed water at room temperature. This storage environment ensured the preservation of the required properties for the acoustic coupling medium.

**Validation experiments for evaluation of LIPUS power effects.** To validate LIPUS power for in vivo studies, a submersible hydrophone and preamplifier were placed in degassed water beneath the coverslip and PVA cone to evaluate the delivered power of LIPUS treatment using the paradigm designed in previous sections. The distance between the transducer and hydrophone was set to 14 mm to allow enough space for the PVA cone (Fig. 1a). Attenuation of ultrasound power was assessed in both the $X-Y$ plane (Top right figure in Fig. 1a) and $Y-Z$ plane (bottom right figure in Fig. 1a). Degree of attenuation was quantified by comparing the power at the various locations to the power at the center of $X-Y$ plane, specifically at the putative brain surface beneath the coverslip. Amplifier voltage was adjusted, 135 V peak-to-peak, to achieve $I_{SPTA}$ 300 mW/cm$^2$ accounting for acoustic signal attenuation from propagation through the acoustic couplant, coverslip, and water. To estimate the potential thermal effect of LIPUS stimulation, a thermocoupler (K-style, connected to an Omega TC-08 sensor) was implanted 1.5 mm deep into the cortex with LIPUS stimulation over one hemisphere. A total of 15 min of LIPUS exposure was performed by maintaining the transducer's supply voltage at a low,

135 V, and high, 210 V, driving voltage divided into three intervals of 5 min each, with a 5 min non-exposure time between each sonication (red and blue periods in Fig. 1b). Temperature change was maintained below 1 °C by using low power, 135 V, sonification paradigm and $I_{SPTA}$ remained below FDA intensity thresholds for low-power ultrasound. These stimulation parameters were carefully selected and verified to ensure that the LIPUS could penetrate the coverslip window, effectively delivering the appropriate level of stimulation to the tissue surrounding the microelectrode implant.

**LIPUS treatment regimen.** Adhering to the parameters and safety measures mentioned above, LIPUS treatment was administered to the transgenic mice on days 0, 1, 2, 3, 4, 5, 6, 7, 14, 21, and 28 above the cranial window (visual cortex). Stimulation was conducted immediately after the first imaging session while in the stereotaxis set up for an exposure time of 15 min (three intervals of 5 min each with 5 min of non-exposure between each sonication). Immediately after, the mouse is transferred back to the two-photon microscope for the second imaging session.

For the rat cohort, LIPUS intensity and duration for each treatment were matched to previously described methods for microscopic investigations of microglial activation (stimulation frequency: 1.13 MHz, burst duration: 22 ms, pulse repetition frequency: 2 Hz, ITA = 300 mW/cm².) Each stimulation session consisted of 3 × 5 min stimulation periods interleaved with 5 min quiescence periods to minimize tissue heating. Subjects received an LIPUS or sham treatment of similar intensity and duration every day for one-week following the electrode implant. For each treatment session, all subjects were lightly anesthetized (2% isoflurane @ 1 L/min in oxygen) and placed in a surgical stereotaxic apparatus prior to mounting the transducer over the electrode. Subjects received an LIPUS or sham treatment of similar intensity and duration every day for one-week following the electrode implant. For the remaining 5 weeks, LIPUS treatments and recording sessions occurred twice a week with each subject being lightly anesthetized during each treatment session.

## Animal preparation

**Mice cohort.** Transgenic CX3CR1-GFP mice (6–8 weeks, strain #:005582, 7 Female/7 Male, N = 7 LIPUS group, N = 7 Control group), which express a green fluorescent protein (GFP) under the CX3CR1 promoter in microglia, were used for this study (Jackson Laboratory, Bar Harbor, ME). Each mouse was implanted with a single shank non-functional microelectrode (NeuroNexus, Sample CM16LP). Post-surgery, mice were housed individually to minimize the risk of damaging their implant or headcap. Subjects were single-housed with environmental enrichment under a 12 h light–dark cycle in a climate-controlled room (20 ± 2.5 °C and relative humidity 30–70%) with continuous access to food and water in accordance with IACUC guidelines.

## Rat cohort

Sprague Dawley rats (3–6 months, 200–300 g, 8 males, Charles River Laboratories, N = 4 LIPUS group, N = 4 Control group) were used for the electrophysiological study and histological analysis. Each rat was implanted with a functional four-shank microelectrode (NeuroNexus, A4x4-5mm-100-177). Post-surgery, the rats were housed individually to minimize the risk of damaging the implant or headcap. Subjects were singly housed with environmental enrichment under a 12 h light–dark cycle in a climate-controlled room (20 ± 2.5 °C and relative humidity 30–70%) with continuous access to food and water in accordance with IACUC guidelines.

## Surgery and probe insertion

**Transgenic mouse surgery.** Surgical methods employed in this study were consistent with previously described experiments[34,38,40,84].

Anesthesia was induced in mice using a drug cocktail consisting of 75 mg/kg ketamine and 7 mg/kg xylazine, administered intraperitoneally. To secure the anesthetized mice throughout the surgery, a stereotaxic frame (Narishige, Amityville, NY) was used. Throughout the procedure, the depth of anesthesia was monitored by observing breathing and toe-pinch responses with additional ketamine doses (40 mg/kg) administered hourly or as needed. The Animal scalps were shaved and thoroughly washed with betadine and ethanol before being removed along with the connective membranes to expose the skull. Small amounts of Vetbond (3 M) were applied to dry the skull surface and enhance the adhesion between the skull and the dental cement headcap. Both the LIPUS-treated mice and control mice underwent chronic surgical procedures. Four bone screws (two over both motor cortices and two over the edges of both visual cortices) were placed and secured to the skull with light-curable dental cement to provide support for a dental cement head cap before craniotomy. A 4 × 4 mm craniotomy was performed, centered over the ipsilateral visual cortex where the microelectrode was implanted. Throughout the drilling process, saline solution was used to clear bone fragments and maintain a cool surgical site to prevent thermal damage to the brain. After the craniotomy, non-functional single-shank silicon probes (NeuroNexus, Sample CM16LP) were inserted at a speed of 200 μm/s through the intact dura mater at a 30° angle parallel to the midline, reaching a depth of ~300 μm below the pial surface (Fig. 1d). Angled insertion was necessary to avoid the probe from colliding with the microscope objective or casting an optical shadow that would preclude imaging the probe tips. Probe positioning on the brain surface was carefully chosen to avoid major blood vessels. Cranial windows were sealed using an in situ curing silicon elastomer (Kwik-Sil, World Precision Instruments) and a glass coverslip, which provided a chronic cranial window for two-photon imaging. Imaging window was then secured to the skull using light-curable dental cement, and a 2-mm-high well was built up around the cranial window to accommodate the water-immersive objective lens used during two-photon imaging. The University of Pittsburgh, Division of Laboratory Animal Resources, and Institutional Animal Care and Use Committee approved all procedures and experimental protocols in strict adherence to the standards for humane animal care as established by the Animal Welfare Act and the National Institutes of Health Guide for the Care and Use of Laboratory Animal.

**Rat surgery.** Subjects (N = 8 Sprague Dawley, 200–300 g, Charles River) were induced with isoflurane (5% concentration @ 2L/min), then received an intraperitoneal injection of ketamine/xylazine/acepromazine (Ketamine: 80-100mg/kg, Xylazine 5-10mg/kg, Acepromazine 1mg/kg) and were maintained on isoflurane for the duration of the procedure (2% concentration @ 1L/min). Subjects were assayed for lack of withdrawal reflex in response to noxious stimuli before being transferred to surgical stereotax (Kopf Instruments, Tujunga, CA) with anesthesia maintained through nose cone (2% @ 1 L/min). A mid-line incision was performed, the scalp resected, and the periosteum removed using a #10 scalpel blade. Two burr holes were formed using a hand-held rotary drill (Dremel, Bosch Tool Corp) and dental burr (#9, Fine Science Tools, Foster City, CA) in the ipsilateral frontal bone and contralateral parietal bone and two #0 titanium bone screws inserted and fixed in place with dental acrylic (C&B Metabond, Parkell, Edgewood, NY). A craniotomy was opened over the right primary whisker barrel cortex with frequent irrigation using cold physiological saline to minimize heating of the pial surface. A durotomy was performed using a 30 g hypodermic needle to expose the cortical surface. Electrodes (A4x4-5mm-100-177 NeuroNexus Technologies Ann Arbor, MI) were implanted perpendicular to the cortical surface using a computer-controlled motorized stereotax with vibration (NeuralGlider Inserter, RRID:SCR_023753; Actuated Medical Inc., Bellefonte, PA) to reduce mechanical compression of pial surface. Ground and reference wires

were attached to existing cranial screws, the craniotomy was sealed with quick-curing silicone elastomer (KwikSil, World Precision Instruments), and all electrodes and cranial screws were bonded together using dental cement (OrthoJet). Following completion of the surgery, subjects received either a sham or LIPUS treatment at the surgical site before being returned to their home cage for recovery.

## Mouse two-photon imaging

A two-photon scanning laser microscope (Ultima IV; Bruker) was employed to capture images of CX3CR1-GFP transgenic mice expressing GFP in microglia (Fig. 1e). The microscope setup included a scan head, an OPO laser (Insight DS+; Spectra-Physics), non-descanned photomultiplier tubes (Hamamatsu), and a ×16, 0.8 numerical aperture water immersion objective lens (Nikon). To enhance vascular contrast, mice were injected intraperitoneally with sulforhodamine 101 (SR101). Microscope laser was set to 920 nm to excite both GFP and SR101, and the resulting fluorescence was recorded in the green and red channels, respectively. Z-stack images and ZT-stack images were acquired at various time points post-implantation, including day 0–7, 14, 21, and 28. Throughout all imaging sessions, mice were securely positioned in the stereotaxic setup. Image stacks covered a horizontal area of 412.8 by 412.8 μm (1024 by 1024 pixels), with depths of ~400 μm. Images were captured above the top shank and/or below the bottom shank, depending on visibility, which could vary due to the presence of blood vessels or surface bleeding on the pial surface. Only microglia located outside the outer shanks and within the same plane as the probes were included in the subsequent analysis. Approximately one hour after the insertion of the probe, the first imaging session was conducted, followed by a session of LIPUS treatment, and the second imaging session. For subsequent imaging sessions, a similar format will be followed: pre-stimulation imaging, stimulation session, post-stimulation imaging.

## Rat electrophysiology recording

Following LIPUS treatment, anesthesia was decreased (1% @ 1 L/min) for 10 min in advance of recording of somatosensory stimulus-evoked multi-unit activity. Electrodes were connected to a NeuroNexus SmartBox Pro through a NeuroNexus headstage, and all data was acquired through NeuroNexus Allego software (bandwidth: 1 Hz–8 kHz, sampling rate: 30 kHz). Stimulation of facial vibrissae was performed using a mechanically actuated, cotton-tipped stimulation arm producing a bi-phasic stimulus (1 Hz, 90° sweep angle stimulating whiskers is protraction and retraction) with stimulation provided to contralateral vibrissae during 3 ×1 min trials with 1-min periods of quiescence between each stimulation interval. Recordings were made on days 0–6, 8, 11, 15, 18, 22, 25, 29, 32, 36, 39, and 43.

## Rat Euthanasia and histological staining of tissue sections

Following final electrophysiology recording sessions, subjects were deeply anesthetized (5% isoflurane @ 2 L/min) and euthanized by transcardial perfusion with a heparinized saline solution (0.9% w/v). Subject tissue was fixed by transcardial perfusion of Phosphate Buffered Saline (PBS) to remove blood followed by 4% paraformaldehyde (PFA) in PBS. Brains were then dissected and allowed to homogenize in a 10% sucrose solution prior to being transferred to a 30% sucrose solution for cryoprotection prior to sectioning.

Brains were embedded in Tissue-Tek OCT compound and 20 μm-thick coronal sections were cut for immunostaining. All sections were washed in 0.1 M phosphate-buffered saline prior to mounting on charged slides (Unifrost Plus, Azer Scientific) for immunostaining to remove section mounting media. Following mounting, sections were washed in a 0.1 M PBS plus 0.05% Triton (Millipore Sigma, X100) solution for 10 min to permeabilize the tissue. Background auto-fluorescence was blocked with a 10-min wash in 0.1 M PBS, 1% Normal Donkey Serum (Abcam, catalog #ab7475). Sections were stained for

Brain Derived Neurotrophic Factor (Abcam, Recombinant Anti-BDNF antibody [EPR1292], catalog #ab108319, Lot #GR3227037-10 (0.271 mg/ml) clonality: monoclonal, 1:100 dilution) and Glial Fibrillary Acidic Protein (Millipore Sigma, Monoclonal Anti-Glial Fibrillary Acidic Protein, catalog #G383, Lot #0000122915 (4–8 mg/mL) clone: G-A-5, clonality: monoclonal, 1:250 dilution) in 0.1 M PBS and allowed to incubate overnight at 4 °C. Following incubation sections were washed 3× for 5 min with 0.1 M PBS to remove excess primary antibody. Primary antibodies were visualized with green secondary (Abcam, Donkey Anti-Rabbit IgG H&L (Alexa Fluor® 488), catalog #ab150073, Lot # GR3313306-1 (2.00 mg/ml), clonality: polyclonal, 1:250 dilution) for BDNF and red secondary (Abcam, Donkey anti-goat IgG H&L Alexa Fluor 594, catalog #ab150132, Lot #GR3290061-3 (2.00 mg/ml), clonality: polyclonal, 1:500 dilution) for GFAP in 0.1 M PBS for 1 h at room temperature. Following incubation, sections were washed 3× for 5 min in 0.1 M PBS to remove excess secondary antibodies. Sections were mounted in a DAPI-fortified mounting solution (Fluorshield, Millipore Sigma, catalog #F6057) to visualize cell nuclei. Slides were covered with a #1 cover glass (VWR, 16004-096) for imaging. Brain sections between 860 and 1100 μm (corresponding to cortical Layer 5) were selected for staining. Sections were imaged on an Olympus Fluoview 1000 microscope.

## Data analysis

The data analysis of images was conducted using ImageJ software (National Institutes of Health)[155]. Specifically, Z-stacks were processed and analyzed to quantify microglial migration, encapsulation of the probes, and the blood vessel diameter. To further analyze temporal characteristics of microglia activity, ZT-stacks were processed and analyzed to quantify microglial activation, surveillance, and density.

**Velocity of migrating microglia.** To track soma migration velocities, we carefully selected images from Z-stacks captured at consecutive time points to maintain consistency in the region of interest. These images were then used to identify the same microglia throughout the time series as follows. First, the 'TurboReg' plugin for ImageJ was used to correct the $x$- and $y$-axis offsets between a pair of images from two consecutive time points (e.g. day 1 and day 2)[156]. These adjusted images were then merged into a single image stack, with the magenta channel representing the earlier time point and the green channel representing the later time point (Fig. 2a). The time interval between these two time points was recorded as D$t$. For each image stack, the shortest distance (D$x$) was measured between the microglia in the magenta channel and their corresponding counterparts in the green channel, with each microglia considered individually. Migration velocity was subsequently calculated using the following formula:

$$\bar{v} = \frac{Dx}{Dt} \tag{1}$$

where $\bar{v}$ represents the average velocity of the microglia between two time points captured in the images. At each time point, we also recorded the minimum distance from the microglia soma to the manually labeled edge of the probe for the microglia's specific location.

**Microglial activation and morphology.** Microglial activation was assessed using a ramification index, which measures the morphological changes in microglia upon activation. When activated, microglia transition from a ramified state to a state with fewer overall processes, but longer processes toward the implantation site[157]. In the image stack, microglia were identified and visually classified as either ramified (1) or transitional (0) microglia in the planes with the probe shank. Distance of these microglia from the surface of the probe was also measured using the 'Measure' feature in ImageJ, and they were

grouped into bins of 50 μm increments up to 300 μm. For each time point, the data was fitted with a logistic regression to determine the probability distribution of microglia being in the ramified or transitional state as a function of the distance from the probe shank[36,38,84]. The suitability of fitting microglial ramification with a logistic regression model is determined by the receiver operating characteristic (ROC) curve[85].

In addition to the ramification index, two other indices were calculated to assess microglial activation: the transitional index (T-index) and the directionality index (D-index). The T-index is based on the length of the longest process extending toward ($n$) and away ($f$) from the probe, while the D-index is derived from the number of processes extending toward ($n$) and away ($f$) from the probe. To determine the direction of process extension, a line parallel to the edge of the probe and passing through the midpoint of each microglial soma was used to distinguish the hemisphere toward and away from the implant. Both indices were calculated at each time point using the following formula[36,38,84]:

$$Index = \frac{(f - n)}{(f + n)} + 1 \qquad (2)$$

Note that this formula restricts the index values to being zero or positive. In cases where there are more and longer microglial projections directed toward the probe, the value of $n$ (representing the length of processes toward the probe) will be greater than $f$ (representing the length of processes away from the probe), resulting in indices approaching zero. On the other hand, when the processes are evenly distributed in terms of both position and length, $n$ and $f$ will be approximately equal, causing the indices to approach one. Therefore, indices of zero indicate activated microglia, while indices of one indicate a ramified state. However, unlike the binary ramification index, the T-index and D-index also provide additional information about the extent of activation from a morphological perspective.

**Microglial surveillance.** Microglial surveillance was quantified based on the previous literature[88]. To create a series of 2D image projections from 0 to 10 min after implantation, grouped z-projections were generated from ZT-stacks with the parameters set to 'Average Intensity' and a group size of 11, transforming the ZT-stacks into T-stacks. To correct for motion, the 'StackReg' plugin for ImageJ was applied to align the T-stacks[156]. Images were thresholded using the 'Li's Minimum Cross Entropy' method[158–160] for better visualization of microglial processes. Microglia surveillance area was measured every minute for 10 min. Areas where the GFP signal appeared or disappeared compared to the previous minute were defined as 'expansion' and 'retraction', respectively. Two datasets were generated from the thresholded datasets: one dataset included 9 consecutive images from the 1st to the 9th minute, and the other dataset included images from the 2nd to the 10th minute. The 'subtraction' operation of the image calculator in ImageJ was used to subtract one dataset from the other and vice versa, resulting in two datasets representing surveillance area expansion and retraction. Average surveillance area expansion and retraction per minute were calculated by counting the average number of pixels over 9 images in each stack. Total surveillance area was defined as the area where the GFP signal (from each microglia) appeared or disappeared at least once within the 10-min period. Total expansion area and total retraction area were created using the Z-projection feature in ImageJ with the projection type set to 'Max Intensity'. By using the 'OR' operation of the image calculator in ImageJ, pixels in either the total expansion area or the total retraction area were included in the total surveillance area (blue region in Fig. 4 bottom row). Stable area was defined as the area where the GFP signal (from a single microglia) remained above the threshold throughout the entire 10-min period. Stable areas (white region in Fig. 4 bottom

row) were generated using the Z-projection feature in ImageJ with the projection type set to 'Min Intensity'.

**Microglial density.** An average of the T-stacks, which were generated in the previous steps, was generated using the 'ZProjection' feature in ImageJ with the projection type set to 'Average Intensity'. Number of microglia was manually counted within the visible regions of the entire field of view (up to 400 μm), excluding the microelectrode region. The density of microglia was then calculated by dividing this number by the area of the visible region, with the microelectrode area excluded.

**Vascular morphology.** Vasculature changes were quantified by measuring blood vessel diameter through a Z-stack projection of the entire blood vessel volume. VasoMetrics ImageJ plugin was used for blood vessel diameter, while a skeletonize ImageJ macro was used to quantify tortuosity and branching number and length[91,92]. The vessel coverage was calculated by the vessel area divided by the total area excluding the microelectrode area.

**Probe coverage.** Microglial encapsulation of the probes was quantified as the percent surface coverage of microglia (GFP) signal[36,161]. Z-stacks were resliced and rotated by 30° using the interactive stack rotation plugin in ImageJ to ensure the entire probe surface was visible in a single frame. The probe surface and the tissue volume up to 20 μm above it were separated into a substack, which was then subjected to a sum projection, resulting in a single projected image. A binary mask of this image was created using the IsoData threshold method in ImageJ[162]. Outline of the probe was manually drawn onto the mask. The probe coverage percentage was calculated by determining the ratio of nonzero pixels representing the threshold GFP signal within the outline to the total number of enclosed pixels using the 'Measure' function.

**Analysis of rat electrophysiology.** Multi-unit spiking activity was analyzed using the open-source Python module SpikeInterface 0.13.0[163], to determine the number of channels actively detecting neural activity during each session as well as signal quality metrics such as electrode impedance, single-unit amplitude, and signal-to-noise ratio[38,51]. Briefly, signals were bandpass filtered between 300 Hz–5 kHz, and a threshold of 3.5 standard deviations above the background was used for action potential detection. Active channels were determined by the detection of at least one identifiable unit during an individual recording session. Action potentials detected across multiple channels were restricted to the channel with the largest amplitude for calculations regarding the number of active channels. Single-unit amplitudes were calculated as the peak-to-peak amplitude of the mean waveform. Signal-to-noise measures were calculated as the peak-to-peak amplitude of the mean waveform for each unit divided by twice the standard deviation of the remaining data. Impedances for each channel were calculated and recorded at 1 kHz prior to each recording session.

**Rat confocal imaging and immunohistochemistry analysis.** Confocal microscope (FluoView 1000, Olympus, Inc., Tokyo, Japan) with ×20 oil-immersive objective lens was used to capture the TIFF images of the probe site and an equivalent area on the contralateral sides. Images were carefully acquired in the resolution of 16-bit (635.9 × 635.9 μm, 1024 × 1024 pixels) with HiLo setting assistance. A previously published MATLAB script (R2023a), I.N.T.E.N.S.I.T.Y. was applied to evaluate the intensity of fluorescent markers (MBP/NG2/MOG/MAP2/NF-200/MCT1/APP) binning away from the probe site[46,134]. Once the probe hole was identified, bins spaced 25 μm apart up to 400 μm away from the probe were generated for DAPI. The average grayscale intensity was calculated as the mean value of all pixels above the threshold of 1.5 standard deviations above the background noise. Cell counting analysis was performed for BDNF and GFAP. Bin size was

modified to 10 μm steps and measured up to 400 μm away from the probe. Cell density was calculated as the total cell counts divided by the tissue area per bin after excluding lost tissue in each bin. Immunohistochemical data was averaged across probe sites and plotted as a function of distance away from the probe.

## Statistical analysis

Statistical analyses were performed using GraphPad Prism software (version 8.0.0 for Windows, GraphPad Software, San Diego, CA, USA, www.graphpad.com). N-values represent individual animals, while n-values represent individual cells. The number of animals per group decreased over time due to death or loss of visibility of the cranial window resulting in images that could not be reliably analyzed. Details about N and n for each figure can be found in the Supplemental Tables (Supplementary Tables 1–7). In most analyses, individual cells were counted as independent data points. Error bars are presented as means ± SEM. A significance level of $\alpha = 0.05$ was used for all analyses. To compare migration, T-index, D-index, surveillance, density, vessel diameter, probe coverage, and IHC fluorescence between animal groups, two-way ANOVAs or mixed-effects models were employed, followed by Holm-Sidak's or Tukey's post hoc comparisons. Welch's t-test was used to evaluate the effect of LIPUS on average vessel diameter on day 0 compared to the control group. Bonferroni correction was applied where appropriate to correct for multiple comparison errors. Logistic regressions were applied to analyze ramification index data, while linear regressions were used to assess the spatial features of migration and surveillance. A linear mixed model that utilized a restricted cubic spline base to fit the nonlinear relationships, was used to analyze electrophysiological data[36].

## Reporting summary

Further information on research design is available in the Nature Portfolio Reporting Summary linked to this article.

## Data availability

All data supporting the findings of this study are available within the article and its supplementary files. Any additional requests for information can be directed to, and will be fulfilled by, the corresponding authors. Source data are provided with this paper.

## Code availability

I.N.T.E.N.S.I.T.Y. Analyzer was published in ref. 46. An updated version can be found at DOI: 10.13140/RG.2.2.14063.65443.

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

## Acknowledgements

This work was supported by NIH R44MH131514, NIH R21EB028055, NIH R01NS094396, NIH R01NS105691, NIH R01NS129632, NIH R01NS115707 and a Diversity Supplement to this parent grant, as well as NSF CAREER 1943906 and an NSF GRFP. The authors would like to acknowledge valuable contributions and feedback on the manuscript from Stieger K, Chen K, Forrest A, Hughes C, Suematsu N, and Zhang G.

## Author contributions

Conceptualization: N.N. Tirko, M. Mulvihill, R. Bagwell, T.D.Y. Kozai. Supervision: N.N. Tirko, J. Greaser, A.S. Alsubhi, K.W. Gheres, T.D.Y. Kozai. Methodology: F. Li, N.N. Tirko, J. Gallego, R. Patel, E. Shaker, G.E.V. Valkenburg, J. Greaser, A.S. Alsubhi, K.W. Gheres, S. Wellman, T.D.Y. Kozai. Investigation: F. Li, J. Gallego, E. Shaker, D. Bashe, N.N. Tirko, A.S. Alsubhi, K.W. Gheres, T.D.Y. Kozai. Formal analysis: F. Li, J. Gallego, R. Patel, E. Shaker, G.E.V. Valkenburg, D. Bashe, V. Singh, K.W. Gheres, C. Padilla, J.I. Broussard, N.N. Tirko, A.S. Alsubhi. Resources: N.N. Tirko, J. Greaser, K.W. Gheres, R. Bagwell, M. Mulvihill, T.D.Y Kozai. Data curation: F. Li, J. Gallego, D. Bashe, R. Patel, E. Shaker, G.E.V. Valkenburg, V. Singh, J.I. Broussard. Visualization: F. Li, J. Gallego, R. Patel, G.E.V. Valkenburg, N.N. Tirko, K.W. Gheres. Writing—original draft: F. Li, J. Gallego. Writing— review & editing: F. Li, J. Gallego, E. Shaker, V. Singh, C. Garcia Padilla, S. Wellman, N.N. Tirko, K.W. Gheres, T.D.Y. Kozai. Project administration: T.D.Y. Kozai, F. Li. Funding acquisition: N.N. Tirko, M. Mulvihill, R. Bagwell, T.D.Y. Kozai.

## Competing interests

J.I. Broussard, N.N. Tirko, J. Greaser, K.W. Gheres, R. Bagwell, and M. Mulvihill have financial stakes in Actuated Medical. Actuated Medical has a patent pending on the hardware design and ultrasound stimulus technology. The inventors are Alanoud S. Alsubhi, Roger B. Bagwell, Jenna K. Greaser, Kevin A. Snook, Natasha A. Tirko, Ryan Clement, and Kyle Gheres. Application # is: 17/837,766. However, it is important to note that the competing interests did not affect the design, methodology, or interpretation of the results in any way. We declare this competing interest to maintain transparency and to provide readers with full disclosure. The remaining authors declare no competing interests.
