## [Peer Review File · Nature Communications]

REVIEWER COMMENTS

Reviewer #1 (Remarks to the Author):

Summary:

Regarding the article titled "Low-Intensity Pulsed Ultrasound Stimulation (LIPUS) Modulates Microglial Activation Following Intracortical Microelectrode Implantation" by Fan Li et al., the authors employed LIPUS intervention to influence the activation and migration of microglia induced by brain-computer interface (BCI) implantation. Using two-photon imaging, the authors observed that LIPUS treatment promoted wound healing in the early phase and reduced microglial encapsulation of the microelectrode, as well as decreased the average vessel diameter on day 28 after BCI probe implantation.

1. In the concluding remarks of Section 3.6, the authors mentioned, 'These findings suggest that LIPUS may mitigate the accumulation of microglia on nearby blood vessels during the transition from the acute phase to the chronic phase.' BCIs may induce brain tissue damage that resembles traumatic brain injury or brain hemorrhage. However, the time point at Day 7 may not adequately represent the chronic phase.

2. The present study, along with prior research, has recognized the impact of LIPUS on the activation level, biological effects, or morphology of glial cells in the brain. The authors employed two-photon imaging to directly observe in vivo changes, consistent with their hypotheses. However, it remains to be investigated whether the intervention of LIPUS can indeed enhance the efficiency of BCIs probes in practice.

3. The astrocytes, similar to microglia, exhibit cell migration characteristics and are more abundant in the central nervous system than microglia. Additionally, they play a role in the formation of the blood-brain barrier. Why was microglia chosen as the focus of this study instead of astrocytes?

4. Endothelial cells within the blood-brain barrier, in conjunction with vascular permeability factor (VPF, VEGF), play a crucial regulatory role in both the repair of brain tissue damage and the management of chronic inflammation. Earlier research has additionally suggested that LIPUS can modulate VEGF, thereby influencing the permeability of the BBB and contributing to the maintenance of dynamic equilibrium. It is recommended to integrate this discussion into the pertinent section of the paper.

5. In future clinical applications, how can LIPUS and BCIs be applied to the human body?

6. In the "2.3. Surgery and Probe Insertion Section"(sentence: 136), a potential typographical error was identified. In Fig. 1b, it should be corrected to Fig 1d

Reviewer #2 (Remarks to the Author):

The manuscript describes an in vivo experiment that investigates the effect of low intensity pulsed ultrasound stimulation (LIPUS) on the host response to implanted brain microelectrodes in mice. This is an area of significant interest in the field of biomaterial host responses, as the host response to implants continues to be a significant challenge and a non-invasive treatment for modulating the inflammatory host response to implants would provide an interesting alternative to material or pharmaceutical-based approaches. The study uses two-photon microscopy to study microglia migration, morphology, and coverage of the implanted electrode over 7 days for LIPUS and control groups, as well as blood vessel size and ratio of microglia associated with blood vessels in the tissue surrounding the implant over 28 days. The most notable results from this work were that LIPUS decreased microglial coverage of the implanted microelectrode from day 6 through 28 compared to the control group, and decreased blood vessel diameter from day 6, although the difference was only statistically significant at day 28. Together, these two long-term observations suggest that LIPUS may decrease neuroinflammation and improve the performance and tissue integration of microelectrodes. Other notable results included that LIPUS modulated microglial migration, with the LIPUS group having increased microglia migration velocity early in the host response (day 1 and 3) and then decreased from day 4-6, compared to the control group. LIPUS groups also resolved to a ramified state earlier than control (day 6 vs day 7; based on morphology at day 6 within 100 μ m of the implant). The LIPUS group also had increased expansion/retraction of microglial on day 7, which was interpreted as microglial surveillance. Limitations of the study are that they focus exclusively on microglia and do not consider blood-derived macrophage and neutrophils that may be present due to local vascular damage and the exclusive use of two-photon microscopy without other methods to validate key findings. Overall, the study provides evidence that LIPUS may lead to faster resolution of the neuroinflammation surrounding brain implants and decrease the impact of the host response on microelectrode performance.

Please note that I do not have expertise in two-photon microscopy and the associated data collection and processing. Therefore, I cannot comment or provide an expert review on the validity of methodology associated with this technique and reserve my comments to data in its presented form.

The authors are commended on the quality of writing. Overall, the manuscript was well-written, and references are appropriate. Please find my specific comments and questions regarding the manuscript below:

Major Comments:

1. The study presents interesting insights into microglial and vascular changes in response to microelectrode implantation and the effect of LIPUS. However, a perceived weakness of the study is that it relies exclusively on two-photon microscopy to assess the host response to the implanted electrode, without additional end-point methods to confirm key results. This approach also limits the potential to explore the effect of LIPUS beyond microglia (visualized with GFP under the control of CX3CR1) and blood vessels (visualized using SR101), and include insights into tissue structure/histology surrounding the electrode, presence of other inflammatory cell types (blood-derived myeloid cells), or proliferation/death of microglia and other cells, etc. Please consider including additional endpoint methods to strengthen conclusions that LIPUS attenuates the host response and improve wound healing in response to brain implants. Conversely, discuss the limitations of TPM in the discussion, and whether the methods used in this manuscript have been validated using other methods.

2. In the methods, the statistical methods seemed appropriate and clear. However, upon reading the supplemental tables, I became confused due to the variability in the number of biological replicates across the different data sets and the number of cells. For many figures, the numbers of animals per group are less than the N=7 reported in the methods, with some as low as N=2. While loss of animals is not unexpected, please clearly indicate why animals were removed from the various data sets. Furthermore, please clearly indicate in the figure legends the number animals used in each figure/panel. I would argue that regardless of the number of cells observed for a particular metric, the number of biological replicates (N) should be greater than two if statistics are to be applied. Were the data (migration velocity, T-index etc.) for each cell treated as an independent data point within the two-way ANOVA or mixed-models? Was there a minimum n from each animal, or could one animal contribute only n=1 or 2, while another contribute n=9 within a data set? These details should be clarified to strengthen impact and robustness of the study.

Minor comments:

3. Line 382-392: The authors note that the suitability of fit (ROC curve) decreased at day 6 and 7 for the LIPUS and control groups, respectively, which were also the times points where significant differences among groups were found. Considering this point, please comment on the validity of the data interpretation for these later time points.

4. How was microglial proliferation accounted for within the study?

5. The LIPUS group was reported to have more rapid migration (although similar metrics of activation) at early time points compared to control. However, the total microglia density (cells/mm²) was not significantly different between the two groups (LIPUS vs Control) over the first 7 days. These observations seem to contradict each other. Can you please clarify what you hypothesize to be happening during the acute phase?

6. The paper focuses exclusively on microglia, and neglect contribution from blood-derived macrophages and neutrophils that are a likely present due to damage to local microvessels and increased permeability of the BBB. Please address this limitation in the discussion.

7. Much of the presented data relies upon manual counting. Were researchers blinded or unblinded when performing counts?

8. Line 64: Please define BCI in main text. (Only defined in Abstract).

9. Line 146 – 150, Section 2.2 Animal Preparation. Please include the sex of the mice used in this study and specify the stock number of the transgenic mice.

10. Line 191 – 192: Please check this sentence “Specifically, Z-stacks were processed and analyzed to quantify microglial migration, encapsulation, and the blood vessel diameter, of the probes.” for typos. I believe it should be “... encapsulation of the probes, and blood vessel diameter”.

11. Methods: The description of the LIPUS treatment regimen for mouse studies was missing from the methods section, although it was discussed in the Results (Figure 1 and paragraph 1 in section 3.1 provides this detail). It would be helpful to include a brief description of the LIPUS methodology within the methods (i.e. applied while animals were in the stereotaxic setup, for a total exposure time of __ minutes, immediately before two-photon imaging?)

12. Figure 3. Microglial Ramification Index. Please review the x-axis of panel A. Text in Results (Line 377-378) report distances up to 300µm, but Day 5, 6 and 7 include distances of up to 400 µm. Please clarify this inconsistency.

13. Ensure each figure caption includes the N, n and clearly states the error bars are SEM.

Reviewer #3 (Remarks to the Author):

Li et al.'s study, which investigates the effects of LIPUS on microglial activation in response to intracortical microelectrode implantation, represents a significant contribution to the field of neuroengineering. The application of two-photon microscopy for an in-depth analysis of microglial dynamics is a commendable aspect of this study. It provides state-of-the-art insights into microglial behavior within the context of FBR, a crucial factor in long-term microelectrode/tissue integration.

Despite these strengths, there are several areas where the study could be improved for a more comprehensive understanding of its implications:

1. Distinction Between Neuroinflammation and FBR: The study primarily focuses on neuroinflammation and microglial activation, which, while valuable, does not fully encapsulate the complexity of FBR. FBR involves a broader range of cellular responses, including systemic immune cells and encapsulation by astrocytes and fibroblasts, leading to collagen deposition and scarring. Unlike traumatic brain injury or degenerative diseases, FBR's unique challenge lies in the persistent presence of a foreign body, necessitating an emphasis on improving the interface just than just healing the tissue.

2. Lack of Histological Analysis for Chronic Effects: The study's evaluation of LIPUS's long-term effects on FBR is limited by the absence of detailed histological analysis at chronic timepoints, such as day 28 or ideally longer durations like 3 months. Assessments should include immunohistochemistry markers for collagen deposition and scarring, such as Alpha SMA staining, to comprehensively evaluate the impact on FBR.

3. Ambiguity in LIPUS Application and Translation: I did not understand the frequency and duration of the LIPUS application from either the method or Figure 1. This raises questions about its practicality and translational potential. It's crucial to clarify whether LIPUS was applied only at surgery or at each time point to understand its feasibility as a treatment and its impact on the

initiation of FBR. The discussion should articulate the translational challenges of delivering LIPUS to humans.

4. Inadequate Exploration of FBR's Complex Nature: The study's limited focus on microglial response, while insightful, overlooks the multifaceted nature of FBR, particularly the roles of systemic immune cells, astrocytes, and fibroblasts in the encapsulation process.

5. Missing Comparative Analysis and Electrophysiological Data: The absence of electrophysiological data and comparative analysis with current anti-FBR treatments (e.g., anti-inflammatory, dexamethasone, or soft coatings) is a notable limitation. Such data are crucial to ascertain the relative effectiveness of LIPUS in mitigating FBR.

6. Generalization in the Discussion Section: The discussion, while informative about neuroinflammation, deviates from the core topic of LIPUS's role in FBR. A more focused discussion on LIPUS's specific impact on FBR would be more relevant and informative for the reader.

7. Lack of Mechanistic Insights: The study hypothesizes about LIPUS's potential mechanisms but lacks experimental data that provide mechanistic insights into how LIPUS influences the cellular and molecular pathways involved in FBR. While this could be part of future studies, the discussion should be more limited (see point 6)

8. Potential Error in Referencing Figures: An inconsistency in referencing Figure 1 within the methods section (lines 161-189) necessitates correction for improved clarity and accuracy.

In summary, while Li et al.'s study offers valuable insights into microglial behavior in FBR through advanced two-photon microscopy, a more comprehensive approach that addresses the complexities of FBR and provides clearer insights into the practical application of LIPUS is essential. The fundamental additional elements required are immunohistochemistry at day 28 and reframing the text from neuro-inflammation to foreign body reaction. Addressing these points would significantly enhance the study's impact and relevance in the context of FBR management in neural implant technologies.

We express our gratitude to the reviewers for their acknowledgement and valuable feedback on this manuscript. Reviewer #1 emphasized, “the authors employed LIPUS intervention to influence the activation and migration of microglia induced by brain-computer interface (BCI) implantation.” Reviewer #2 noted that the manuscript was well-written, and Reviewer #3 highlighted that the application of two-photon microscopy for an in-depth analysis of microglial dynamics is a commendable aspect. Moreover, we highly value their insightful comments on experiment details and data presentations. We have carefully considered and addressed each comment both below (in blue) and in the revised manuscript (presented in purple for easy reference). We believe that these revisions have significantly improved the overall quality of the manuscript.

Reviewer #1: Summary: Regarding the article titled "Low-Intensity Pulsed Ultrasound Stimulation (LIPUS) Modulates Microglial Activation Following Intracortical Microelectrode Implantation" by Fan Li et al., the authors employed LIPUS intervention to influence the activation and migration of microglia induced by brain-computer interface (BCI) implantation. Using two-photon imaging, the authors observed that LIPUS treatment promoted wound healing in the early phase and reduced microglial encapsulation of the microelectrode, as well as decreased the average vessel diameter on day 28 after BCI probe implantation.

1. In the concluding remarks of Section 3.6, the authors mentioned, 'These findings suggest that LIPUS may mitigate the accumulation of microglia on nearby blood vessels during the transition from the acute phase to the chronic phase.' BCIs may induce brain tissue damage that resembles traumatic brain injury or brain hemorrhage. However, the time point at Day 7 may not adequately represent the chronic phase.

RESPONSE 1: We took this into consideration and made the appropriate changes.

Results 3.6 “These findings suggest that LIPUS may mitigate the accumulation of microglia on nearby blood vessels during the **inflammatory** phase.”

2. The present study, along with prior research, has recognized the impact of LIPUS on the activation level, biological effects, or morphology of glial cells in the brain. The authors employed two-photon imaging to directly observe in vivo changes, consistent with their hypotheses. However, it remains to be investigated whether the intervention of LIPUS can indeed enhance the efficiency of BCIs probes in practice.

RESPONSE 2: It can be reasoned that if we are able to mediate the inflammatory response with LIPUS, especially when looking at the difference in microglial probe coverage between the two groups, as this has a direct effect on the recording performance of BCI's probes. However, to address this reviewer's specific concerns, we have added electrophysiological results on recording performance.

“LIPUS improved neuronal SU activity and decreased astrocytic scarring

To examine the influence of LIPUS on the functionality of the implanted microelectrode, electrophysiological data was recorded from L5 of lightly anesthetized rats through somatosensory stimuli. We examined the single-unit (SU) recording between the LIPUS and Control groups independent of laminar depth and averaged all the channels. SU yield was calculated as a percentage of channels along the shank that detected at least one SU. SU Yield ($32.81 \pm 16.01\%$) in the LIPUS group was significantly higher than in the control ($10.94 \pm 10.94\%$) at the end of the 6-week study (Fig. 8a). Both groups experienced a rapid decline in SU Yield in the first two post-operative weeks. LIPUS-treated group exhibited a stabilization at around day 29 while the control continued to exhibit a decline, which can also be observed in the number of active channels (Fig. 8b). Similarly, the SU signal-to-noise ratio (SNR), a measurement of the strength of SU activity, showed significant improvements between day 14-21 and at day 43 (Fig. 8c). Also, SU amplitude was consistently higher in the LIPUS group and significantly greater between 5-6 weeks (Fig. 8d). Average SU SNR in the LIPUS group was 6.56 ± 3.32 on day 43 compared to control at 0.533 ± 0.533 . The average amplitude in the LIPUS group on day at the end of 6 weeks was $155.31 \pm 41.33 \mu\text{V}$ compared to $10.69 \pm 10.68 \mu\text{V}$ for control. Noise floor remained relatively consistent in both groups, with the LIPUS group having a slightly higher noise profile during the first week (Fig. 8e, Day 0: 9.75 ± 0.95 vs $5.85 \pm 1.11 \mu\text{V}$). There was no significant difference in device impedance between the LIPUS and control groups during the 6-week period (Fig. 8f, $p = 0.2936$). Together, these results suggest that LIPUS may increase the recording performance of L5 neural activity at later time points.”

“Figure 8. LIPUS improves neuronal single-unit activity and decreases astrocytic scar formation around the probe. a) Single-unit yield over time. **b)** Number of active channels over time. **c)** Single-unit SNR. **d)** Single-unit amplitude. **e)** Noise floor. **f)** Average impedance. * indicates significant group wise differences via a linear mixed model likelihood ratio test with a 95% confidence interval. **g)** Left image: Representative histological stain for GFAP around a microelectrode probe hole in LIPUS-treated and control animals; Middle: GFAP intensity if increased in the control group compared to the LIPUS treated group 30 μm away from the probe; Right: violin plot quantifies GFAP within the first 50 μm away from the shank (Sidak’s multiple comparison, \wedge $p < 0.0001$, * $p < 0.0371$; Welch’s t-test **** $p < 0.0001$). **h)** Left image: Representative histological stain for BDNF around a microelectrode probe hole in LIPUS-treated and control animals; Middle: BDNF staining showed no difference between the LIPUS and Control groups; Right: violin plot quantifies BDNF within 50 μm away from the shank. **i)** Representative histological stain for DAPI, BDNF, and GFAP. a-f: LIPUS N = 4; Control N = 4. g-h: LIPUS N = 3, n = 10; Control N = 3, n = 9. Scale bar = 100 μm . Error bars indicate the S.E.M.”

3. The astrocytes, similar to microglia, exhibit cell migration characteristics and are more abundant in the central nervous system than microglia. Additionally, they play a role in the formation of the blood-brain barrier. Why were microglia chosen as the focus of this study instead of astrocytes?

RESPONSE 3: Microglia cells were chosen since our studies as well as others in the field have identified that microglia are the first responders to implants, significantly changing their morphology and activity within seconds. In contrast, we showed that astrocytes processes move slower (order of hours) and astrocyte soma do not migrate (Savya et al *Biomaterials* 2022). Therefore, we reasoned that microglia would yield more sensitive metrics for quantification over the earlier period of the treatment. Furthermore, our lab is familiar with the morphology and behavior within the context of neuroinflammation, probe insertion, and various mediation methods, as supported by six papers published about microglia cells since 2012. We also hope that this paper lays the foundation for future studies investigating neurons, oligodendrocytes, blood vessels, and astrocytes during LIPUS as well as conducting electrophysiological recordings and behavioral studies. However, as the reviewer notes, astrocytes undergo hypertrophy and upregulation of GFAP hours after injury, especially as evaluated by IHC. Therefore, we conduct GFAP IHC as additional data to support the manuscript (as seen above). We expand this rationale in the introduction:

Introduction Line 44-77 “Penetrating microelectrode arrays that interface with the nervous system are front-end components of Brain-computer interfaces (BCI), which have demonstrated remarkable potential for restoring

motor and sensory function¹⁶⁻¹⁸. One key challenge is the complex FBR caused by neuroinflammation after insertion of microelectrodes. Neuroinflammation is a multifaceted process orchestrated through interactions of blood cells¹⁹, endothelial cells²⁰, and glial cells²¹, particularly microglia²²⁻²⁵. These microglial processes can often be detrimental to intracortical microelectrode interfaces, which are designed to detect neuronal signals and study neural activity²²⁻²⁵. The implantation of a foreign body such as a microelectrode into the brain disrupts the BBB^{26,27}, degenerates neurons^{15,28,29} and oligodendrocytes^{15,30-32}, and activates microglia³³⁻³⁷, astrocytes^{33-35,38-43}, and NG2 glia^{40,42,43}. Among glial cells, microglia are first responders and important mediators of neuroinflammation, protecting the brain from injury^{39,44}. Microglia's filipodia, or processes, enable them to efficiently survey the surrounding area, detecting pathogens, disturbances, or foreign bodies^{39,45}. Previous studies show that microglia direct their processes toward the microelectrode minutes after implantation, followed by astrocyte processes on the order of hours and NG2 glial processes on the order of days^{35,40,41,46,47}. During this phase, microglia tend to extend both longer and greater number of processes toward the injury site while reducing the length and number of processes away from the injury site³⁹. Next, microglia begin to migrate toward microelectrodes within 12 – 24 hours, followed by NG2 glia^{35,40,41,46,47}. By contrast, astrocytes do not migrate, instead they swell and become hypertrophic⁴⁰. Over weeks, these glial cells form the glial scar surrounding the implanted probe⁴⁸, resulting in the characteristic FBR^{35,40,41,47}.

This FBR adds an insulating layer on the microelectrode⁴⁹ increasing the impedance⁵⁰ and the recorded noise floor⁵¹ of the microelectrode. Further, persistent microglial activation upregulates production of proinflammatory cytokines⁵²⁻⁵⁴ contributing to progressive neurodegeneration, reducing the number of recorded neurons surrounding the microelectrode⁵⁵, and decreasing the number of detectable single-units⁵⁶. Additionally, proinflammatory microglia can attach to blood vessels, initiate upregulation of proinflammatory profiles, and phagocytose astrocyte endfeet, breaking down the neurovascular unit⁵⁷. Phagocytic microglia also contribute to the loss of neurons and synapses via complement activation^{58,59}, disrupting the neural circuit and adversely affecting the propagation of neural signals. Moreover, proinflammatory microglia promote the release of nitric oxide (NO)⁶⁰, causing abnormal dilation of the cerebral vessels⁶¹. Together with other glia cells, such as astrocytes, NG2 glial cells and oligodendrocytes^{40,41,62,63}, this neuroinflammatory response increases the noise in electrophysiological recordings, decreases the strength of the neural signal being recorded, and obstructs the tissue-microelectrode integration. Although administration of drugs such as dexamethasone³⁸ and HOE-642³³ or coating microelectrodes with zwitterionic polymer³⁴ and neuroadhesive L1³⁵ reduces microglial activation, these interventions require either recurrent injections or complex manufacturing processes⁶⁴. Therefore, modulating microglial changes following microelectrode implantation remains a challenge in neuroscience research and clinical BCI applications.”

4. Endothelial cells within the blood-brain barrier, in conjunction with vascular permeability factor (VPF, VEGF), play a crucial regulatory role in both the repair of brain tissue damage and the management of chronic inflammation. Earlier research has additionally suggested that LIPUS can modulate VEGF, thereby influencing the permeability of the BBB and contributing to the maintenance of dynamic equilibrium. It is recommended to integrate this discussion into the pertinent section of the paper.

RESPONSE 4: We took this into consideration and added the below to the discussion.

Discussion Lines 472-475 “Recent studies have shown that LIPUS, in a similar manner to activated microglia, can drive **VEGF expression**, a vascular permeability factor that supports vascular formation¹⁶. Then as blood vessels are being repaired and fibrin and cell debris are cleared, microglia near the microelectrode start to transition back to a ramified state suggesting a resolution of inflammation (Fig. 4).”

5. In future clinical applications, how can LIPUS and BCIs be applied to the human body?

RESPONSE 5: We took this into consideration and added the below to the discussion to address both the limitations of this study in that regard and how the two could be investigated in the future.

Discussion Lines 479-507 “A few limitations exist within the study due to the restrictions of the experimental design and inherent data collected. First, a major limitation is the lack of detailed histological analysis at multiple depths and a wider breadth of staining. Future studies should examine fibroblasts, neurons, oligodendrocytes, astrocytes, and other cells involved in the FBR in greater detail, in doing so would determine cellular and molecular pathways that LIPUS may be modulating. Furthermore, more extensive histology would allow for the observation of microglia proliferation as we were unable to do so within this study. Although we performed staining for BDNF and GFAP, they were only performed at a single depth (approximately cortical L5). A future

study will need to be conducted to further characterize the attenuation of LIPUS over distance, as many neurological diseases and disorders not only affect the cortex but deep brain structures.

An important future direction is how to translate LIPUS for BCI. An examination of how LIPUS would penetrate the tissue through the skulls of non-human primates and humans would be beneficial as the results will be widely different than in rodents. Additionally, examining whether LIPUS will work together with deep brain stimulation or single element and array transducers, will expand the usage of LIPUS within the clinical space. For deeper implants, focused ultrasound or LIPUS through an array of ultrasound transducers could be used to focus ultrasound waves to deep brain targets, such as for deep brain stimulation electrodes, without overstimulating shallower brain regions¹⁶³.

To optimize the application of ultrasound stimulation for both the bio-integration of BCIs and the treatment of neurodegenerative diseases, future studies should explore the parameter space of ultrasound stimulation to achieve optimal therapeutic benefits. These parameters include intensity¹⁶⁴, frequency¹⁶⁵, pulse repetition frequency¹⁶⁶, and duty cycle¹⁶⁷. Among these parameters, it was observed that higher intensity ultrasound led to an increase in blood flow¹⁶⁸. However, excessively high intensity could potentially breach the BBB¹⁶⁹. Furthermore, different frequencies of ultrasound have the potential to induce distinct biological responses. For example, lower frequencies (e.g. 350 kHz) activate TRPA1 channels, while higher frequencies (43 MHz) stimulate piezo1 channels¹⁶⁵. Moreover, the adjustment of pulse repetition has the potential to selectively modulate excitatory and inhibitory neurons¹⁶⁶, providing a promising approach to precise neuromodulation. In addition to parameter considerations, the variations in bone density within the skull pose challenges for precise targeting of LIPUS to the intended treatment area¹⁷⁰. Future studies should focus on refining the models of ultrasound propagation through the skull¹⁷¹ and developing phase arrays¹⁷² to steer focal ultrasound to improve the precision of LIPUS treatment.”

6. In the “2.3. Surgery and Probe Insertion Section”(sentence: 136), a potential typographical error was identified. In Fig. 1b, it should be corrected to Fig 1d

RESPONSE 6: We have checked the references and have fixed them in the manuscript.

Methods 5.3 “reaching a depth of approximately 300 μm below the pial surface (**Fig. 1d**).”

Methods 5.4 “A two-photon scanning laser microscope (Ultima IV; Bruker) was employed to capture images of CX3CR1-GFP transgenic mice expressing GFP in microglia (**Fig. 1e**).”

Reviewer #2: The manuscript describes an in vivo experiment that investigates the effect of low intensity pulsed ultrasound stimulation (LIPUS) on the host response to implanted brain microelectrodes in mice. This is an area of significant interest in the field of biomaterial host responses, as the host response to implants continues to be a significant challenge and a non-invasive treatment for modulating the inflammatory host response to implants would provide an interesting alternative to material or pharmaceutical-based approaches. The study uses two-photon microscopy to study microglia migration, morphology, and coverage of the implanted electrode over 7 days for LIPUS and control groups, as well as blood vessel size and ratio of microglia associated with blood vessels in the tissue surrounding the implant over 28 days. The most notable results from this work were that LIPUS decreased microglial coverage of the implanted microelectrode from day 6 through 28 compared to the control group, and decreased blood vessel diameter from day 6, although the difference was only statistically significant at day 28. Together, these two long-term observations suggest that LIPUS may decrease neuroinflammation and improve the performance and tissue integration of microelectrodes. Other notable results included that LIPUS modulated microglial migration, with the LIPUS group having increased microglia migration velocity early in the host response (day 1 and 3) and then decreased from day 4-6, compared to the control group. LIPUS groups also resolved to a ramified state earlier than control (day 6 vs day 7; based on morphology at day 6 within 100 μm of the implant). The LIPUS group also had increased expansion/retraction of microglial on day 7, which was interpreted as microglial surveillance. Limitations of the study are that they focus exclusively on microglia and do not consider blood-derived macrophage and neutrophils that may be present due to local vascular damage and the exclusive use of two-photon microscopy without other methods to validate key findings. Overall, the study provides evidence that LIPUS may lead to faster resolution of the neuroinflammation surrounding brain implants and decrease the impact of the host response on microelectrode performance. Please note that I do not have expertise in two-photon microscopy and the associated data collection and processing. Therefore, I cannot comment or provide an expert review on the validity of methodology associated with this technique and reserve my comments to data in its presented form. The authors are commended on the quality of writing. Overall, the manuscript was

well-written, and references are appropriate. Please find my specific comments and questions regarding the manuscript below:

1. The study presents interesting insights into microglial and vascular changes in response to microelectrode implantation and the effect of LIPUS. However, a perceived weakness of the study is that it relies exclusively on two-photon microscopy to assess the host response to the implanted electrode, without additional end-point methods to confirm key results. This approach also limits the potential to explore the effect of LIPUS beyond microglia (visualized with GFP under the control of CX3CR1) and blood vessels (visualized using SR101), and include insights into tissue structure/histology surrounding the electrode, presence of other inflammatory cell types (blood-derived myeloid cells), or proliferation/death of microglia and other cells, etc. Please consider including additional endpoint methods to strengthen conclusions that LIPUS attenuates the host response and improve wound healing in response to brain implants. Conversely, discuss the limitations of TPM in the discussion, and whether the methods used in this manuscript have been validated using other methods.

RESPONSE 1: We understand the reviewer's comments regarding the limitations of our study by only viewing microglia cells and blood vessels via two-photon imaging. As pointed out, the limited nature of our mouse model and the imaging did not let use investigations these other aspects. However, in the newly added IHC figure (as seen below), we are able to show BDNF and GFAP around the electrode 6 weeks after implantation in rats. Additionally, exploring these cell types in different mouse models could be a potential future area of investigation. We have added this point to the Discussion.

Results 3.8 "To determine whether LIPUS modulated astrogliosis, immunohistochemical analysis was performed 6 weeks after implantation. Astrogliosis was examined by glial fibrillary acidic protein (GFAP) labeling LIPUS significantly reduced intensity compared to the control group within the 50 μm from the probe (Fig. 8g, 1.64 ± 0.081 vs. $2.59 \pm 0.14 \mu\text{m}$). This lower activation of astrocyte cells around the probe further validates the lessening glial scarring and could explain some of the improvements seen in the electrophysiological data. Because brain-derived neurotrophic factor (BDNF) has been implicated as a potential mechanism related to LIPUS, we examined BDNF activity around the implant. The fluorescent intensity of BDNF showed no difference in the two groups (Fig. 8h). LIPUS showed increased recording performance in L5 neurons as well as reduced astrocyte FBR around chronically implanted arrays."

“Figure 8. LIPUS improves neuronal single-unit activity and decreases astrocytic scar formation around the probe. a) Single-unit yield over time. b) Number of active channels over time. c) Single-unit SNR. d) Single-unit amplitude. e) Noise floor. f) Average impedance. * indicates significant group wise differences via a linear

mixed model likelihood ratio test with a 95% confidence interval. **g)** Left image: Representative histological stain for GFAP around a microelectrode probe hole in LIPUS-treated and control animals; Middle: GFAP intensity if increased in the control group compared to the LIPUS treated group 30 μm away from the probe; Right: violin plot quantifies GFAP within the first 50 μm away from the shank (Sidak's multiple comparison, $^{\wedge} p < 0.0001$, $^* p < 0.0371$; Welch's t-test $^{****} p < 0.0001$). **h)** Left image: Representative histological stain for BDNF around a microelectrode probe hole in LIPUS-treated and control animals; Middle: BDNF staining showed no difference between the LIPUS and Control groups; Right: violin plot quantifies BDNF within 50 μm away from the shank. **i)** Representative histological stain for DAPI, BDNF, and GFAP. a-f: LIPUS N = 4; Control N = 4. g-h: LIPUS N = 3, n = 10; Control N = 3, n = 9. Scale bar = 100 μm . Error bars indicate the S.E.M."

Discussion Lines 479-487 "A few limitations exist within the study due to the restrictions of the experimental design and inherent data collected. First, a major limitation is the lack of detailed histological analysis at multiple depths and a wider breadth of staining. Future studies should examine fibroblasts, neurons, oligodendrocytes, astrocytes, and other cells involved in the FBR in greater detail, in doing so would determine cellular and molecular pathways that LIPUS may be modulating. Furthermore, more extensive histology would allow for the observation of microglia proliferation as we were unable to do so within this study. Although we performed staining for BDNF and GFAP, they were only performed at a single depth (approximately cortical L5). A future study will need to be conducted to further characterize the attenuation of LIPUS over distance, as many neurological diseases and disorders not only affect the cortex but deep brain structures."

2. In the methods, the statistical methods seemed appropriate and clear. However, upon reading the supplemental tables, I became confused due to the variability in the number of biological replicates across the different data sets and the number of cells. For many figures, the numbers of animals per group are less than the N=7 reported in the methods, with some as low as N=2. While loss of animals is not unexpected, please clearly indicate why animals were removed from the various data sets. Furthermore, please clearly indicate in the figure legends the number animals used in each figure/panel. I would argue that regardless of the number of cells observed for a particular metric, the number of biological replicates (N) should be greater than two if statistics are to be applied. Were the data (migration velocity, T-index etc.) for each cell treated as an independent data point within the two-way ANOVA or mixed-models? Was there a minimum n from each animal, or could one animal contribute only n=1 or 2, while another contribute n=9 within a data set? These details should be clarified to strengthen the impact and robustness of the study.

RESPONSE 2: The number of animals per group decreased over time due to death or loss of visibility in the glass window resulting in images that could not be reliably analyzed. Typically, in many of the analyses, each cell was counted as independent data points, which we have clarified in the methods 2.8 statistics section. Additionally N and n for each group and analysis were added to each figure caption. This is also expanded upon in the supplementary section, in which we made to clarify N and n on every day for all analysis. Regarding the analysis for the extension/retraction/surveillance (where one group only had N=2), we used a statistical method that account for this discrepancy, so even with the addition to animal that group, there would still be significance. The lowest number of cells an animal could contribute was 3 cells; but in most of our analysis (barring figure 4), each animal contributed more than 3 cells. For figure 7, most animals contributed 1 or 2 blood vessels due to the image quality and difficulties tracking the same blood vessel during all timepoints. N for number of animals was added to all figure captions.

Methods 5.8 "N-values represent individual animals, while n-values represent individual cells. The number of animals per group decreased over time due to death or loss of visibility of the cranial window resulting in images that could not be reliably analyzed. Details about N and n for each figure can be found in the Supplemental Tables. In most analysis, individual cells were counted as independent data points."

3. Line 382-392: The authors note that the suitability of fit (ROC curve) decreased at day 6 and 7 for the LIPUS and control groups, respectively, which were also the times points where significant differences among groups were found. Considering this point, please comment on the validity of the data interpretation for these later time points.

RESPONSE 3: The suitability of fit (ROC curve) was used to determine the fit of only the microglial ramification. The T-index and D-index are different measurements in which the T-index is based on the length of the longest process extending towards and away from the probe while the D-index is based on the number of processes extending towards and away from the probe. We clarify this by changing the figure so that the ramification and the ROC curves are classified as panel (a) together and made this clarification in the text...

Results 3.3 Line 206-208 “ROC curves were created by plotting the true positive rate against false positive rate for various threshold settings used to classify observations as positive or negative (detailed calculation documented here⁹⁰) and was applied to analyze the **ramification plots.**”

4. How was microglial proliferation accounted for within the study?

RESPONSE 4: We did not specifically account for microglia proliferation, especially for later time points where microglial cells could potentially migrate faster than can be resolved by our imaging time points. However, we quantify the spatial distribution of activation, which provides the degree of morphological microglial activation as a function of distance from the focal injury and FBR. Our significant findings provide valuable information on microglial activation and the effect of LIPUS on cortical implants. We clarify in the discussion that we did not specifically study proliferation and that future studies could examine the role of LIPUS on microglial proliferation.

Discussion Line 479-487 “A few limitations exist within the study due to the restrictions of the experimental design and inherent data collected. First, a major limitation is the lack of detailed histological analysis at multiple depths and a wider breadth of staining. Future studies should examine fibroblasts, neurons, oligodendrocytes, astrocytes, and other cells involved in the FBR in greater detail, in doing so would determine cellular and molecular pathways that LIPUS may be modulating. **Furthermore, more extensive histology would allow for the observation of microglia proliferation as we were unable to do so within this study.** Although we performed staining for BDNF and GFAP, they were only performed at a single depth (approximately cortical L5). A future study will need to be conducted to further characterize the attenuation of LIPUS over distance, as many neurological diseases and disorders not only affect the cortex but deep brain structures.”

5. The LIPUS group was reported to have more rapid migration (although similar metrics of activation) at early time points compared to control. However, the total microglia density (cells/mm²) was not significantly different between the two groups (LIPUS vs Control) over the first 7 days. These observations seem to contradict each other. Can you please clarify what you hypothesize to be happening during the acute phase?

RESPONSE 5: Density was calculated over the whole field of view, so the density fluctuated slightly, as microglia are both migrating towards and away from the probe, but not enough for significance testing.

Furthermore, some researchers have observed that microglia associated with vasculature can migrate along the vessels^{52,53}. We clarify in the methods section how the density was calculated.

Methods 5.7.4 “An average of the T-stacks, which were generated in the previous steps, was generated using the 'ZProjection' feature in ImageJ with the projection type set to 'Average Intensity'. The number of microglia was manually counted within the visible regions **of the entire field of view (up to 400 μm)**, excluding the microelectrode region. The density of microglia was then calculated by dividing this number by the area of the visible region, with the microelectrode area excluded.”

Discussion Line 412-414 “Despite the increased migratory speed of microglia, our results did not show a change in microglia density between the two groups over time, as observations were limited to a 400 μm area and therefore **changes in bulk density were not tracked** similar to previous two-photon studies³⁹.”

6. The paper focuses exclusively on microglia, and neglect contribution from blood-derived macrophages and neutrophils that are a likely present due to damage to local microvessels and increased permeability of the BBB. Please address this limitation in the discussion.

RESPONSE 6: We have considered these comments and have now added the lines below.

Discussion Lines 475-478 “**Blood-derived macrophages and neutrophils** also aid in the resolution of inflammation as they can clear pathogens and increase the permeability of the BBB^{136,137}. Future studies would investigate more in-depth how blood vessel permeability and macrophages are affected by LIPUS treatment.”

7. Much of the presented data relies upon manual counting. Were researchers blinded or unblinded when performing counts?

RESPONSE 7: All analysis had some level of manual and automated processing, and the researchers were not blinded. The surgeon conducted two surgeries a day and was blind to which animal would be stimulated. The person who imaged the animals immediately after randomly generated which animal would undergo stimulation. Researchers conducted the analysis and the leads on the project, Fan Li and Jazlyn Gallego, checked the work by doing random samples or doing all of it themselves.

8. Line 64: Please define BCI in main text. (Only defined in Abstract).

RESPONSE 8: We have made the change to the manuscript.

Introduction Lines 44-46 “Penetrating microelectrode arrays that interface with the nervous system are front-end components of **Brain-computer interfaces (BCI)**, which have demonstrated remarkable potential for restoring motor and sensory function¹⁶⁻¹⁸.”

9. Line 146 – 150, Section 2.2 Animal Preparation. Please include the sex of the mice used in this study and specify the stock number of the transgenic mice.

RESPONSE 9: We have considered this suggestion and have now included this additional information in the manuscript. The sex of the mice and stock number have been added.

Methods 5.2.1 “Transgenic CX3CR1-GFP mice (6-8 weeks, **strain #:005582, 7 Female/7 Male**, N = 7 LIPUS group, N = 7 control group), which express green fluorescent protein (GFP) under the CX3CR1 promoter in microglia, were used for this study (Jackson Laboratory, Bar Harbor, ME).”

10. Line 191 – 192: Please check this sentence “Specifically, Z-stacks were processed and analyzed to quantify microglial migration, encapsulation, and the blood vessel diameter, of the probes.” for typos. I believe it should be “... encapsulation of the probes, and blood vessel diameter”.

RESPONSE 10: We have made the change to the manuscript.

Methods 5.7 “Specifically, Z-stacks were processed and analyzed to quantify microglial migration, **encapsulation of the probes, and the blood vessel diameter.**”

11. Methods: The description of the LIPUS treatment regimen for mouse studies was missing from the methods section, although it was discussed in the Results (Figure 1 and paragraph 1 in section 3.1 provides this detail). It would be helpful to include a brief description of the LIPUS methodology within the methods (i.e. applied while animals were in the stereotaxic setup, for a total exposure time of ___ minutes, immediately before two-photon imaging?)

RESPONSE 11: We have considered this suggestion and have now included this additional information in the manuscript.

Methods 5.1.4 “**5.1.4 LIPUS Treatment Regimen**

Adhering to parameters and safety measures mentioned above, LIPUS treatment was administered to the transgenic mice on days 0, 1, 2, 3, 4, 5, 6, 7, 14, 21, 28 above the cranial window (visual cortex). Stimulation was conducted **immediately after the first imaging session** while in the stereotaxis set up for an **exposure time of 15 min (three intervals of 5 min each with 5 min of non-exposure between each sonication)**. Immediately after, the mouse is transferred back to the two-photon microscope for the **second imaging session.**”

12. Figure 3. Microglial Ramification Index. Please review the x-axis of panel A. Text in Results (Line 377-378) report distances up to 300µm, but Day 5, 6 and 7 include distances of up to 400 µm. Please clarify this inconsistency.

RESPONSE 12: We have found the inconsistency and have fixed it within the text.

Results 3.3 “Microglia were sampled within a range of 0 to **400 µm** from the probe and fitted into a logistic regression model (Fig. 3a). The Y-axis represents the percentage of ramified microglia at each distance bin (50: 0-50 µm, 100: 50-100 µm, 150: 100-150 µm, 200: 150-200 µm, 250: 200-250 µm, 300: 250-300 µm, **350: 300-350 µm, 400: 350-400 µm**).”

13. Ensure each figure caption includes the N, n and clearly states the error bars are SEM.

RESPONSE 12: We have added N, n, and a clear statement of error bars = SEM when necessary to each figure.

We have edited all figure captions to clearly state N, n, and SEM.

Reviewer #3: Li et al.'s study, which investigates the effects of LIPUS on microglial activation in response to intracortical microelectrode implantation, represents a significant contribution to the field of neuroengineering. The application of two-photon microscopy for an in-depth analysis of microglial dynamics is a commendable aspect of this study. It provides state-of-the-art insights into microglial behavior within the context of FBR, a crucial factor in long-term microelectrode/tissue integration. Despite these strengths, there are several areas where the study could be improved for a more comprehensive understanding of its implications:

1. Distinction Between Neuroinflammation and FBR: The study primarily focuses on neuroinflammation and microglial activation, which, while valuable, does not fully encapsulate the complexity of FBR. FBR involves a broader range of cellular responses, including systemic immune cells and encapsulation by astrocytes and fibroblasts, leading to collagen deposition and scarring. Unlike traumatic brain injury or degenerative diseases, FBR's unique challenge lies in the persistent presence of a foreign body, necessitating an emphasis on improving the interface just than just healing the tissue.

RESPONSE 1: The reviewer makes a valid point in that the foreign body response is more complex and challenging than simply neuroinflammation. We have revised the introduction and discussion to emphasize the FBR.

Introduction Lines 46-48, 56-63 “One key challenge is the complex FBR caused by neuroinflammation after insertion of microelectrodes. Neuroinflammation is a multifaceted process orchestrated through interactions of blood cells¹⁹, endothelial cells²⁰, and glial cells²¹, particularly microglia²²⁻²⁵...During this phase, microglia tend to extend both longer and greater number of processes toward the injury site while reducing the length and number of processes away from the injury site³⁹. Next, microglia begin to migrate toward microelectrodes within 12 – 24 hours, followed by NG2 glia^{35,40,41,46,47}. By contrast, astrocytes do not migrate, instead they swell and become hypertrophic⁴⁰. Over weeks, these glial cells form the glial scar surrounding the implanted probe⁴⁸, resulting in the characteristic FBR^{35,40,41,47}.”

This FBR adds an insulating layer on the microelectrode⁴⁹ increasing the impedance⁵⁰ and the recorded noise floor⁵¹ of the microelectrode...”

Discussion Lines 427-432, 439-440 “If the FBR and inflammation are not resolved timely, events such as oxidative stress¹²² and microglia priming¹²³ can lead to persistent activation of microglia. As a result, activated microglia secrete proinflammatory cytokines that lead to the formation of a glial scar⁵⁴. The glial scar not only displaces neurons from the recording range^{54,124} but also forms a physical barrier, inhibiting ion diffusion and preventing the propagation and detection of neural signals^{17,52,86,125,126}. This encapsulation sheath, a hallmark of the foreign body response, is made up of microglia cells, astrocytes, fibroblasts, and extracellular build-up^{127,128}... The FBR involves several signaling pathways that can be regulated by mechanosensitive and thermosensitive channels^{144,145}.”

2. Lack of Histological Analysis for Chronic Effects: The study's evaluation of LIPUS's long-term effects on FBR is limited by the absence of detailed histological analysis at chronic timepoints, such as day 28 or ideally longer durations like 3 months. Assessments should include immunohistochemistry markers for collagen deposition and scarring, such as Alpha SMA staining, to comprehensively evaluate the impact on FBR.

RESPONSE 2: We recognize the value of incorporating representative immunobiological images at a later timepoint and assessing markers to evaluate the impact on the foreign body response. To address this, we quantified DAPI, BDNF, and GFAP in Figure 8 in rats after 6 weeks (more specifically 43 days).

Results 3.8 “To determine whether LIPUS modulated astrogliosis, immunohistochemical analysis was performed 6 weeks after implantation. Astrogliosis was examined by glial fibrillary acidic protein (GFAP) labeling LIPUS significantly reduced intensity compared to the control group within the 50 μm from the probe (Fig. 8g, 1.64 ± 0.081 vs. $2.59 \pm 0.14 \mu\text{m}$). This lower activation of astrocyte cells around the probe further validates the lessening glial scarring and could explain some of the improvements seen in the electrophysiological data. Because brain-derived neurotrophic factor (BDNF) has been implicated as a potential mechanism related to LIPUS, we examined BDNF activity around the implant. The fluorescent intensity of BDNF showed no difference in the two groups (Fig. 8h). LIPUS showed increased recording performance in L5 neurons as well as reduced astrocyte FBR around chronically implanted arrays.”

“Figure 8. LIPUS improves neuronal single-unit activity and decreases astrocytic scar formation around the probe. a) Single-unit yield over time. **b)** Number of active channels over time. **c)** Single-unit SNR. **d)** Single-unit amplitude. **e)** Noise floor. **f)** Average impedance. * indicates significant group wise differences via a linear mixed model likelihood ratio test with a 95% confidence interval. **g)** Left image: Representative histological stain for GFAP around a microelectrode probe hole in LIPUS-treated and control animals; Middle: GFAP intensity if increased in the control group compared to the LIPUS treated group 30 μm away from the probe; Right: violin plot quantifies GFAP within the first 50 μm away from the shank (Sidak’s multiple comparison, \wedge $p < 0.0001$, * $p < 0.0371$; Welch’s t-test **** $p < 0.0001$). **h)** Left image: Representative histological stain for BDNF around a microelectrode probe hole in LIPUS-treated and control animals; Middle: BDNF staining showed no difference between the LIPUS and Control groups; Right: violin plot quantifies BDNF within 50 μm away from the shank. **i)** Representative histological stain for DAPI, BDNF, and GFAP. a-f: LIPUS N = 4; Control N = 4. g-h: LIPUS N = 3, n = 10; Control N = 3, n = 9. Scale bar = 100 μm. Error bars indicate the S.E.M.”

3. Ambiguity in LIPUS Application and Translation: I did not understand the frequency and duration of the LIPUS application from either the method or Figure 1. This raises questions about its practicality and translational potential. It’s crucial to clarify whether LIPUS was applied only at surgery or at each time point to understand its feasibility as a treatment and its impact on the initiation of FBR. The discussion should articulate the translational challenges of delivering LIPUS to humans.

RESPONSE 3: We have considered the feedback and made clarifications to the methods section to address these concerns. Due to the limitations of the equipment, we had to image and stimulate separately (image under two-photon, move to stimulation set-up, move back to two-photon for more imaging). Furthermore, the surgery was conducted in a conjoined surgical room and not under the two-photon. This is something we are actively working on with Actuated Medical as they are currently building and testing new transducers that will allow for simultaneous imaging and stimulation.

Methods Section 5.1.4 “5.1.4 LIPUS Treatment Regimen

Adhering to parameters and safety measures above, LIPUS treatment was administered on days 0, 1, 2, 3, 4, 5, 6, 7, 14, 21, 28 above the cranial window (visual cortex). **Stimulation was conducted immediately after the first imaging session while in the stereotaxis setup for an exposure time of 15 minutes (three intervals of 5 minutes each with 5 minutes of non-exposure between each sonication. Immediately after, the mouse is transferred back to the two-photon for the second imaging session.”**

Methods Section 5.4 “Approximately **one hour after the insertion of the probe**, the first imaging session was conducted, followed by a session of LIPUS treatment, and the second imaging session. For subsequent imaging sessions, a similar format will be followed: **pre-stimulation imaging, stimulation session, post-stimulation imaging.**”

4. Inadequate Exploration of FBR's Complex Nature: The study's limited focus on microglial response, while insightful, overlooks the multifaceted nature of FBR, particularly the roles of systemic immune cells, astrocytes, and fibroblasts in the encapsulation process.

RESPONSE 4: We understand the reviewer's comments regarding other important players in the formation of glial scars and how the complex nature of FBR has been inadequately studied for brain implants. In this context, our goal is not to characterize the complex nature of the FBR but to demonstrate that LIPUS can be applied to implants to reduce the FBR and enhance performance. However, in response to the reviewer, we added IHC studies that quantified astrocytes and BDNF around the electrode 6 weeks after implantation in rats in addition to recording performance. We hope that these studies future inspire novel investigations into the complex nature of FBR using novel interventions such as LIPUS. We have added this point to the Discussion. Discussion Line 479-487 “A few limitations exist within the study due to the restrictions of the experimental design and inherent data collected. First, a major limitation is the lack of detailed histological analysis at multiple depths and a wider breadth of staining. Future studies should examine fibroblasts, neurons, oligodendrocytes, astrocytes, and other cells involved in the FBR in greater detail, in doing so would determine cellular and molecular pathways that LIPUS may be modulating. Furthermore, more extensive histology would allow for the observation of microglia proliferation as we were unable to do so within this study. Although we performed staining for BDNF and GFAP, they were only performed at a single depth (approximately cortical L5). A future study will need to be conducted to further characterize the attenuation of LIPUS over distance, as many neurological diseases and disorders not only affect the cortex but deep brain structures.”

5. Missing Comparative Analysis and Electrophysiological Data: The absence of electrophysiological data and comparative analysis with current anti-FBR treatments (e.g., anti-inflammatory, dexamethasone, or soft coatings) is a notable limitation. Such data are crucial to ascertain the relative effectiveness of LIPUS in mitigating FBR.

RESPONSE 5: In response to the reviewer, we have added electrophysiological data. The study is well controlled with comparative analysis between untreated and LIPUS treatment. LIPUS treatment of cortical electrodes has not been investigated before and the results presented here demonstrate that LIPUS may be a intervention that improves cortical implants. We hope that this work inspires additional comparative research with current anti-FBR treatments, especially in comparison to system treatments. Notably, other Nature Communications or Nature family publications on novel technologies and interventions do not include multifaceted comparative analysis (only novel design/intervention vs control device). In a similar vein, we believe that this work presents a novel enough discovery without a multifaceted comparative analysis on multiple different interventions. See:

Jiang, S., Patel, D.C., Kim, J. *et al.* Spatially expandable fiber-based probes as a multifunctional deep brain interface. *Nat Commun* **11**, 6115 (2020).

Chen, J.C., Bhave, G., Alrashdan, F. *et al.* Self-rectifying magnetoelectric metamaterials for remote neural stimulation and motor function restoration. *Nat. Mater.* **23**, 139–146 (2024).

Kozai, T., Langhals, N., Patel, P. *et al.* Ultrasmall implantable composite microelectrodes with bioactive surfaces for chronic neural interfaces. *Nature Mater* **11**, 1065–1073 (2012).

Hunt, D.L., Lai, C., Smith, R.D. *et al.* Multimodal in vivo brain electrophysiology with integrated glass microelectrodes. *Nat Biomed Eng* **3**, 741–753 (2019).

Zhang, L., Cao, Z., Bai, T. *et al.* Zwitterionic hydrogels implanted in mice resist the foreign-body reaction. *Nat Biotechnol* **31**, 553–556 (2013).

Discussion Line 449-451 “Additionally, future studies should investigate the performance of LIPUS compared to other well-established intervention strategies, such as dexamethasone⁶⁸.”

RESPONSE 5: To address this reviewer's specific concerns, we have added electrophysiological results on recording performance in rats over the course of 6 weeks. Within this analysis we observe the SU yield, SU SNR, SU amplitude, noise floor, and impedance (as provided below). Additionally, 6 weeks were chosen as a previous rat study show unit recordings stabilize at 6 weeks^{69,70}.

Results 3.8 “To examine the influence of LIPUS on the functionality of the implanted microelectrode, electrophysiological data was recorded from L5 of lightly anesthetized rats through somatosensory stimuli. We

examined the single-unit (SU) recording between the LIPUS and Control groups independent of laminar depth and averaged all the channels. SU yield was calculated as a percentage of channels along the shank that detected at least one SU. SU Yield ($32.81 \pm 16.01\%$) in the LIPUS group was significantly higher than in the control ($10.94 \pm 10.94\%$) at the end of the 6-week study (Fig. 8a). Both groups experienced a rapid decline in SU Yield in the first two post-operative weeks. LIPUS-treated group exhibited a stabilization at around day 29 while the control continued to exhibit a decline, which can also be observed in the number of active channels (Fig. 8b). Similarly, the SU signal-to-noise ratio (SNR), a measurement of the strength of SU activity, showed significant improvements between day 14-21 and at day 43 (Fig. 8c). Also, SU amplitude was consistently higher in the LIPUS group and significantly greater between 5-6 weeks (Fig. 8d). Average SU SNR in the LIPUS group was 6.56 ± 3.32 on day 43 compared to control at 0.533 ± 0.533 . The average amplitude in the LIPUS group on day at the end of 6 weeks was $155.31 \pm 41.33 \mu\text{V}$ compared to $10.69 \pm 10.68 \mu\text{V}$ for control. Noise floor remained relatively consistent in both groups, with the LIPUS group having a slightly higher noise profile during the first week (Fig. 8e, Day 0: 9.75 ± 0.95 vs $5.85 \pm 1.11 \mu\text{V}$). There was no significant difference in device impedance between the LIPUS and control groups during the 6-week period (Fig. 8f, $p = 0.2936$). Together, these results suggest that LIPUS may increase the recording performance of L5 neural activity at later time points.”

“Figure 8. LIPUS improves neuronal single-unit activity and decreases astrocytic scar formation around the probe. a) Single-unit yield over time. **b)** Number of active channels over time. **c)** Single-unit SNR. **d)** Single-unit amplitude. **e)** Noise floor. **f)** Average impedance. * indicates significant group wise differences via a linear mixed model likelihood ratio test with a 95% confidence interval. **g)** Left image: Representative histological stain for GFAP around a microelectrode probe hole in LIPUS-treated and control animals; Middle: GFAP intensity if increased in the control group compared to the LIPUS treated group 30 µm away from the probe; Right: violin plot quantifies GFAP within the first 50 µm away from the shank (Sidak’s multiple comparison, $^{\wedge} p < 0.0001$, $^* p < 0.0371$; Welch’s t-test $^{****} p < 0.0001$). **h)** Left image: Representative histological stain for BDNF around a microelectrode probe hole in LIPUS-treated and control animals; Middle: BDNF staining showed no difference between the LIPUS and Control groups; Right: violin plot quantifies BDNF within 50 µm away from the shank. **i)** Representative histological stain for DAPI, BDNF, and GFAP. a-f: LIPUS N = 4; Control N = 4. g-h: LIPUS N = 3, n = 10; Control N = 3, n = 9. Scale bar = 100 µm. Error bars indicate the S.E.M.”

6. Generalization in the Discussion Section: The discussion, while informative about neuroinflammation, deviates from the core topic of LIPUS's role in FBR. A more focused discussion on LIPUS's specific impact on FBR would be more relevant and informative for the reader.

RESPONSE 6: We understand that within the discussion we need to be more selective about our word choice as to not confuse or misinform the readers, so we have made some changes to make the focus more on the foreign body response as seen from some excerpts below. As well as split the discussion into sections with the titles focused on the FBR.

Discussion Lines 392-394, 427-428, 439-440 “The impact of LIPUS on improving long-term electrophysiological recordings was investigated 6 weeks after the implantation of a 4-shank intracortical microelectrode. Previous studies have indicated that unit recordings stabilize around this timeframe, which could be due to the late-onset phase of the FBR^{99,100}... If the FBR and inflammation are not resolved timely, events such as oxidative stress¹¹⁶ and microglia priming¹¹⁷ can lead to persistent activation of microglia... The FBR involves several signaling pathways that can be regulated by mechanosensitive and thermosensitive channels^{138,139}.”

7. Lack of Mechanistic Insights: The study hypothesizes about LIPUS's potential mechanisms but lacks experimental data that provide mechanistic insights into how LIPUS influences the cellular and molecular pathways involved in FBR. While this could be part of future studies, the discussion should be more limited (see point 6)

RESPONSE 7: We have limited speculation within the discussion about potential mechanism and clarified that this is a limitation within our study and could be an avenue for future research.

Discussion Line 479-487 “A few limitations exist within the study due to the restrictions of the experimental design and inherent data collected. First, a major limitation is the lack of detailed histological analysis at multiple depths and a wider breadth of staining. Future studies should examine fibroblasts, neurons, oligodendrocytes, astrocytes, and other cells involved in the FBR in greater detail, in doing so would determine cellular and molecular pathways that LIPUS may be modulating. Furthermore, more extensive histology would allow for the observation of microglia proliferation as we were unable to do so within this study. Although we performed staining for BDNF and GFAP, they were only performed at a single depth (approximately cortical L5). A future study will need to be conducted to further characterize the attenuation of LIPUS over distance, as many neurological diseases and disorders not only affect the cortex but deep brain structures.”

8. Potential Error in Referencing Figures: An inconsistency in referencing Figure 1 within the methods section (lines 161-189) necessitates correction for improved clarity and accuracy.

RESPONSE 8: We have checked the references and have fixed them in the manuscript.

Methods 5.3.1 “reaching a depth of approximately 300 μm below the pial surface (**Fig. 1d**).”

Methods 5.4 “A two-photon scanning laser microscope (Ultima IV; Bruker) was employed to capture images of CX3CR1-GFP transgenic mice expressing GFP in microglia (**Fig. 1e**).”

In summary, while Li et al.'s study offers valuable insights into microglial behavior in FBR through advanced two-photon microscopy, a more comprehensive approach that addresses the complexities of FBR and provides clearer insights into the practical application of LIPUS is essential. The fundamental additional elements required are immunohistochemistry on day 28 and reframing the text from neuro-inflammation to foreign body reaction. Addressing these points would significantly enhance the study's impact and relevance in the context of FBR management in neural implant technologies.

- 1 Yu, K., Niu, X., Krook-Magnuson, E. & He, B. Intrinsic functional neuron-type selectivity of transcranial focused ultrasound neuromodulation. *Nature communications* **12**, 2519 (2021).
- 2 Niu, X., Yu, K. & He, B. Transcranial focused ultrasound induces sustained synaptic plasticity in rat hippocampus. *Brain stimulation* **15**, 352-359 (2022).
- 3 Kamimura, H. A., Conti, A., Toschi, N. & Konofagou, E. E. Ultrasound neuromodulation: Mechanisms and the potential of multimodal stimulation for neuronal function assessment. *Frontiers in physics* **8**, 150 (2020).
- 4 Wang, Q. *et al.* Low-intensity pulsed ultrasound attenuates postoperative neurocognitive impairment and salvages hippocampal synaptogenesis in aged mice. *Brain Sciences* **13**, 657 (2023).
- 5 Jiang, X. *et al.* A review of low-intensity pulsed ultrasound for therapeutic applications. *IEEE Transactions on Biomedical Engineering* **66**, 2704-2718 (2018).

- 6 Dell'Italia, J., Sanguinetti, J. L., Monti, M. M., Bystritsky, A. & Reggente, N. Current state of potential mechanisms supporting low intensity focused ultrasound for neuromodulation. *Frontiers in human neuroscience* **16**, 872639 (2022).
- 7 Chu, Y.-C., Lim, J., Chien, A., Chen, C.-C. & Wang, J.-L. Activation of mechanosensitive ion channels by ultrasound. *Ultrasound in Medicine & Biology* **48**, 1981-1994 (2022).
- 8 Qiu, Z. *et al.* The mechanosensitive ion channel Piezo1 significantly mediates in vitro ultrasonic stimulation of neurons. *IScience* **21**, 448-457 (2019).
- 9 Hsu, C. H., Pan, Y. J., Zheng, Y. T., Lo, R. Y. & Yang, F. Y. Ultrasound reduces inflammation by modulating M1/M2 polarization of microglia through STAT1/STAT6/PPAR γ signaling pathways. *CNS Neuroscience & Therapeutics* (2023).
- 10 Lai, S.-W. *et al.* Regulatory effects of neuroinflammatory responses through brain-derived neurotrophic factor signaling in microglial cells. *Molecular Neurobiology* **55**, 7487-7499 (2018).
- 11 Chen, T.-T., Lan, T.-H. & Yang, F.-Y. Low-intensity pulsed ultrasound attenuates LPS-induced neuroinflammation and memory impairment by modulation of TLR4/NF- κ B signaling and CREB/BDNF expression. *Cerebral Cortex* **29**, 1430-1438 (2019).
- 12 Su, W.-S., Wu, C.-H., Song, W.-S., Chen, S.-F. & Yang, F.-Y. Low-intensity pulsed ultrasound ameliorates glia-mediated inflammation and neuronal damage in experimental intracerebral hemorrhage conditions. *Journal of Translational Medicine* **21**, 565 (2023).
- 13 Chen, R. *et al.* Protective effects of low-intensity pulsed ultrasound (LIPUS) against cerebral ischemic stroke in mice by promoting brain vascular remodeling via the inhibition of ROCK1/p-MLC2 signaling pathway. *Cerebral Cortex*, bhad330 (2023).
- 14 dos Santos Tramontin, N. *et al.* Effects of Low-Intensity Transcranial Pulsed Ultrasound Treatment in a Model of Alzheimer's Disease. *Ultrasound in Medicine & Biology* **47**, 2646-2656 (2021).
- 15 Kaloss, A. M. *et al.* Noninvasive Low-Intensity Focused Ultrasound Mediates Tissue Protection following Ischemic Stroke. *BME Frontiers* **2022** (2022).
- 16 Ichijo, S. *et al.* Low-intensity pulsed ultrasound therapy promotes recovery from stroke by enhancing angio-neurogenesis in mice in vivo. *Scientific reports* **11**, 4958 (2021).
- 17 Zheng, T., Du, J., Yuan, Y. & Liu, L. Neuroprotective effect of low-intensity transcranial ultrasound stimulation in moderate traumatic brain injury rats. *Frontiers in Neuroscience* **14**, 506541 (2020).
- 18 Su, W.-S., Wu, C.-H., Chen, S.-F. & Yang, F.-Y. Low-intensity pulsed ultrasound improves behavioral and histological outcomes after experimental traumatic brain injury. *Scientific reports* **7**, 15524 (2017).
- 19 Rayasam, A., Fukuzaki, Y. & Vexler, Z. S. Microglia-leucocyte axis in cerebral ischaemia and inflammation in the developing brain. *Acta Physiologica* **233**, e13674 (2021).
- 20 Wu, F. *et al.* CXCR2 is essential for cerebral endothelial activation and leukocyte recruitment during neuroinflammation. *Journal of neuroinflammation* **12**, 1-15 (2015).
- 21 Yang, Q. q. & Zhou, J. w. Neuroinflammation in the central nervous system: Symphony of glial cells. *Glia* **67**, 1017-1035 (2019).
- 22 Meneghetti, N. *et al.* Narrow and broad gamma bands process complementary visual information in mouse primary visual cortex. *Eneuro* (2021).
- 23 Mizuseki, K., Royer, S., Diba, K. & Buzsáki, G. Activity dynamics and behavioral correlates of CA3 and CA1 hippocampal pyramidal neurons. *Hippocampus* **22**, 1659-1680 (2012).
- 24 Huber, D. *et al.* Multiple dynamic representations in the motor cortex during sensorimotor learning. *Nature* **484**, 473-478 (2012).
- 25 Jia, X., Smith, M. A. & Kohn, A. Stimulus selectivity and spatial coherence of gamma components of the local field potential. *Journal of Neuroscience* **31**, 9390-9403 (2011).
- 26 Flesher, S. N. *et al.* A brain-computer interface that evokes tactile sensations improves robotic arm control. *Science* **372**, 831-836 (2021).
- 27 Collinger, J. L. *et al.* High-performance neuroprosthetic control by an individual with tetraplegia. *Lancet* **381**, 557-564 (2013). [https://doi.org/10.1016/S0140-6736\(12\)61816-9](https://doi.org/10.1016/S0140-6736(12)61816-9)
- 28 Hochberg, L. R. *et al.* Reach and grasp by people with tetraplegia using a neurally controlled robotic arm. *Nature* **485**, 372-375 (2012). <https://doi.org/10.1038/nature11076>
- 29 Kozai, T. D. Y. *et al.* Reduction of neurovascular damage resulting from microelectrode insertion into the cerebral cortex using in vivo two-photon mapping. *Journal of neural engineering* **7**, 046011 (2010).
- 30 Wellman, S. M., Li, L., Yaxiaer, Y., McNamara, I. N. & Kozai, T. D. Revealing spatial and temporal patterns of cell death, glial proliferation, and blood-brain barrier dysfunction around implanted intracortical neural interfaces. *Frontiers in Neuroscience* **13**, 493 (2019).

- 31 Michelson, N. J. *et al.* Multi-scale, multi-modal analysis uncovers complex relationship at the brain tissue-implant neural interface: new emphasis on the biological interface. *Journal of neural engineering* **15**, 033001 (2018).
- 32 Eles, J. R., Vazquez, A. L., Kozai, T. D. & Cui, X. T. In vivo imaging of neuronal calcium during electrode implantation: spatial and temporal mapping of damage and recovery. *Biomaterials* **174**, 79-94 (2018).
- 33 Savva, S. P. *et al.* In vivo spatiotemporal dynamics of astrocyte reactivity following neural electrode implantation. *Biomaterials* **289**, 121784 (2022).
- 34 Wellman, S. M. & Kozai, T. D. In vivo spatiotemporal dynamics of NG2 glia activity caused by neural electrode implantation. *Biomaterials* **164**, 121-133 (2018).
- 35 Chen, K., Wellman, S. M., Yaxiaer, Y., Eles, J. R. & Kozai, T. D. In vivo spatiotemporal patterns of oligodendrocyte and myelin damage at the neural electrode interface. *Biomaterials* **268**, 120526 (2021).
- 36 Wellman, S. M., Li, L., Yaxiaer, Y., McNamara, I. & Kozai, T. D. Revealing spatial and temporal patterns of cell death, glial proliferation, and blood-brain barrier dysfunction around implanted intracortical neural interfaces. *Frontiers in neuroscience* **13**, 493 (2019).
- 37 Dubaniewicz, M. *et al.* Inhibition of Na⁺/H⁺ exchanger modulates microglial activation and scar formation following microelectrode implantation. *Journal of neural engineering* **18**, 045001 (2021).
- 38 Yang, Q. *et al.* Zwitterionic polymer coating suppresses microglial encapsulation to neural implants in vitro and in vivo. *Advanced biosystems* **4**, 1900287 (2020).
- 39 Eles, J. R. *et al.* Neuroadhesive L1 coating attenuates acute microglial attachment to neural electrodes as revealed by live two-photon microscopy. *Biomaterials* **113**, 279-292 (2017).
- 40 Kozai, T. D., Jaquins-Gerstl, A. S., Vazquez, A. L., Michael, A. C. & Cui, X. T. Dexamethasone retrodialysis attenuates microglial response to implanted probes in vivo. *Biomaterials* **87**, 157-169 (2016).
- 41 Kozai, T. D. Y., Vazquez, A. L., Weaver, C. L., Kim, S.-G. & Cui, X. T. In vivo two-photon microscopy reveals immediate microglial reaction to implantation of microelectrode through extension of processes. *Journal of neural engineering* **9**, 066001 (2012).
- 42 Chen, K., Padilla, C. G., Kiselyov, K. & Kozai, T. D. Cell-specific alterations in autophagy-lysosomal activity near the chronically implanted microelectrodes. *Biomaterials* **302**, 122316 (2023).
- 43 Szarowski, D. *et al.* Brain responses to micro-machined silicon devices. *Brain research* **983**, 23-35 (2003).
- 44 Li, Q. & Barres, B. A. Microglia and macrophages in brain homeostasis and disease. *Nature Reviews Immunology* **18**, 225-242 (2018).
- 45 Bernier, L.-P. *et al.* Nanoscale surveillance of the brain by microglia via cAMP-regulated filopodia. *Cell reports* **27**, 2895-2908. e2894 (2019).
- 46 Sharon, A., Jankowski, M. M., Shmoel, N., Erez, H. & Spira, M. E. Inflammatory foreign body response induced by neuro-implants in rat cortices depleted of resident microglia by a CSF1R inhibitor and its implications. *Frontiers in neuroscience* **15**, 646914 (2021).
- 47 Kozai, T. D. *et al.* Chronic tissue response to carboxymethyl cellulose based dissolvable insertion needle for ultra-small neural probes. *Biomaterials* **35**, 9255-9268 (2014).
- 48 Li, T. *et al.* Proliferation of parenchymal microglia is the main source of microgliosis after ischaemic stroke. *Brain* **136**, 3578-3588 (2013).
- 49 Tao, L. *et al.* Microglia modulation with 1070-nm light attenuates A β burden and cognitive impairment in Alzheimer's disease mouse model. *Light: Science & Applications* **10**, 179 (2021).
- 50 Folloni, D. *et al.* Manipulation of subcortical and deep cortical activity in the primate brain using transcranial focused ultrasound stimulation. *Neuron* **101**, 1109-1116. e1105 (2019).
- 51 Nahm, F. S. Receiver operating characteristic curve: overview and practical use for clinicians. *Korean journal of anesthesiology* **75**, 25-36 (2022).
- 52 Grossmann, R. *et al.* Juxtavascular microglia migrate along brain microvessels following activation during early postnatal development. *Glia* **37**, 229-240 (2002).
- 53 Checchin, D., Sennlaub, F., Levavasseur, E., Leduc, M. & Chemtob, S. Potential role of microglia in retinal blood vessel formation. *Invest Ophthalmol Vis Sci* **47**, 3595-3602 (2006).
<https://doi.org/10.1167/iovs.05-1522>
- 54 Krukiewicz, K. Electrochemical impedance spectroscopy as a versatile tool for the characterization of neural tissue: A mini review. *Electrochemistry Communications* **116**, 106742 (2020).

- 55 Frampton, J. P., Hynd, M. R., Shuler, M. L. & Shain, W. Effects of glial cells on electrode impedance recorded from neural prosthetic devices in vitro. *Annals of biomedical engineering* **38**, 1031-1047 (2010).
- 56 Woeppel, K., Dhawan, V., Shi, D. & Cui, X. T. Nanotopography-enhanced biomimetic coating maintains bioactivity after weeks of dry storage and improves chronic neural recording. *Biomaterials* **302**, 122326 (2023).
- 57 Kozai, T. D., Jaquins-Gerstl, A. S., Vazquez, A. L., Michael, A. C. & Cui, X. T. Brain tissue responses to neural implants impact signal sensitivity and intervention strategies. *ACS chemical neuroscience* **6**, 48-67 (2015).
- 58 Perry, V. H. & Teeling, J. in *Seminars in immunopathology*. 601-612 (Springer).
- 59 Salatino, J. W., Ludwig, K. A., Kozai, T. D. Y. & Purcell, E. K. Glial responses to implanted electrodes in the brain. *Nature BME* (2017).
- 60 Loane, D. J., Kumar, A., Stoica, B. A., Cabatbat, R. & Faden, A. I. Progressive neurodegeneration after experimental brain trauma: association with chronic microglial activation. *Journal of Neuropathology & Experimental Neurology* **73**, 14-29 (2014).
- 61 Fiáth, R. *et al.* Slow insertion of silicon probes improves the quality of acute neuronal recordings. *Scientific Reports* **9**, 111 (2019).
- 62 Haruwaka, K. *et al.* Dual microglia effects on blood brain barrier permeability induced by systemic inflammation. *Nature communications* **10**, 5816 (2019).
- 63 Butler, C. A. *et al.* Microglial phagocytosis of neurons in neurodegeneration, and its regulation. *Journal of neurochemistry* **158**, 621-639 (2021).
- 64 Wilton, D. K. *et al.* Microglia and complement mediate early corticostriatal synapse loss and cognitive dysfunction in Huntington's disease. *Nature Medicine*, 1-19 (2023).
- 65 Goodwin, J. L., Uemura, E. & Cunnick, J. E. Microglial release of nitric oxide by the synergistic action of β -amyloid and IFN- γ . *Brain research* **692**, 207-214 (1995).
- 66 Van Mil, A. H. *et al.* Nitric oxide mediates hypoxia-induced cerebral vasodilation in humans. *Journal of applied physiology* **92**, 962-966 (2002).
- 67 Wellman, S. M. *et al.* A materials roadmap to functional neural interface design. *Advanced functional materials* **28**, 1701269 (2018).
- 68 Kozai, T. D. Y., Jaquins-Gerstl, A. S., Vazquez, A. L., Michael, A. C. & Cui, X. T. Dexamethasone retrodialysis attenuates microglial response to implanted probes in vivo. *Biomaterials* **87**, 157-169 (2016). <https://doi.org/10.1016/j.biomaterials.2016.02.013>
- 69 Ludwig, K. A., Uram, J. D., Yang, J., Martin, D. C. & Kipke, D. R. Chronic neural recordings using silicon microelectrode arrays electrochemically deposited with a poly (3, 4-ethylenedioxythiophene)(PEDOT) film. *Journal of neural engineering* **3**, 59 (2006).
- 70 Liu, X. *et al.* Stability of the interface between neural tissue and chronically implanted intracortical microelectrodes. *IEEE transactions on rehabilitation engineering* **7**, 315-326 (1999).
- 71 Chen, K., Cambi, F. & Kozai, T. D. Pro-myelinating Clemastine administration improves recording performance of chronically implanted microelectrodes and nearby neuronal health. *Biomaterials* **301**, 122210 (2023).
- 72 Wellman, S. M. *et al.* Cuprizone-induced oligodendrocyte loss and demyelination impairs recording performance of chronically implanted neural interfaces. *Biomaterials* **239**, 119842 (2020).
- 73 Chang, J.-W., Wu, M.-T., Song, W.-S. & Yang, F.-Y. Ultrasound stimulation suppresses LPS-induced proinflammatory responses by regulating NF- κ B and CREB activation in microglial cells. *Cerebral Cortex* **30**, 4597-4606 (2020).
- 74 Qin, L., Liu, Y., Hong, J. S. & Crews, F. T. NADPH oxidase and aging drive microglial activation, oxidative stress, and dopaminergic neurodegeneration following systemic LPS administration. *Glia* **61**, 855-868 (2013).
- 75 Neher, J. J. & Cunningham, C. Priming microglia for innate immune memory in the brain. *Trends in Immunology* **40**, 358-374 (2019).
- 76 Kang, W. & Hébert, J. M. Signaling pathways in reactive astrocytes, a genetic perspective. *Mol Neurobiol* **43**, 147-154 (2011). <https://doi.org/10.1007/s12035-011-8163-7>

REVIEWERS' COMMENTS

Reviewer #1 (Remarks to the Author):

The issues have been all addressed.

Reviewer #2 (Remarks to the Author):

Dear Authors and Editors,

Thank you for providing the revised manuscript and the opportunity to review the manuscript entitled "Low-Intensity Pulsed Ultrasound Stimulation (LIPUS)

Modulates Microglial Activation Following Intracortical Microelectrode Implantation" by Fan Li et al.". The authors have now addressed my critiques and concerns adequately, and I have no further comments.

Reviewer #3 (Remarks to the Author):

Happy with the changes made. No further comments.